# Gated Integration of Low-Rank Adaptation for Continual Learning of Large Language Models

**Yan-Shuo Liang, Jia-Rui Chen and Wu-Jun Li**[*]
National Key Laboratory for Novel Software Technology,
School of Computer Science, Nanjing University, P. R. China
{liangys,jiaruichen2005}@smail.nju.edu.cn, liwujun@nju.edu.cn

## Abstract

Continual learning (CL), which requires the model to learn multiple tasks sequentially, is crucial for large language models (LLMs). Recently, low-rank adaptation (LoRA), one of the most representative parameter-efficient fine-tuning (PEFT) methods, has gained increasing attention in CL of LLMs. However, most existing CL methods based on LoRA typically expand a new LoRA branch to learn each new task and force the new and old LoRA branches to influence old tasks equally, potentially leading to forgetting. In this work, we propose a new method, called gated integration of low-rank adaptation (GainLoRA), for CL of LLMs. GainLoRA expands a new LoRA branch for each new task and introduces gating modules to integrate the new and old LoRA branches. Furthermore, GainLoRA leverages the new gating module to minimize the influence from the new LoRA branch to old tasks, effectively mitigating forgetting and improving the model's overall performance. Experimental results on CL benchmarks demonstrate that GainLoRA outperforms existing state-of-the-art methods. Code is available at https://github.com/liangyanshuo/gainlora.

## 1 Introduction

Continual learning (CL), which requires the model to learn multiple tasks sequentially, is crucial for large language models (LLMs) [48]. Specifically, although existing LLMs have demonstrated strong performance for a wide range of tasks [4, 8, 53, 54, 73] with extensive pre-trained knowledge and further fine-tuning strategies, they may lose knowledge acquired from old tasks when learning multiple tasks sequentially. This phenomenon, known as catastrophic forgetting [36, 41, 59, 61], highlights the need for developing effective CL methods for LLMs. Existing CL methods can be categorized into two main categories. The first category [45] assumes that task identities are available during inference, while the second category [32, 75] tackles a more difficult and practical setting where task identities are unavailable during inference.

Recently, low-rank adaptation (LoRA) [21], one of the most representative parameter-efficient fine-tuning (PEFT) methods, has gained increasing attention in the CL of LLMs [3, 61]. Specifically, by reparameterizing pre-trained weights in a low-rank form, LoRA updates only a limited number of parameters to adapt LLMs to a downstream task, making the fine-tuning process much more efficient than updating all parameters of LLMs [17]. This efficiency also benefits CL, making LoRA increasingly popular in CL of LLMs.

Most existing CL methods based on LoRA [32, 75] typically expand a new LoRA branch for learning each new task while freezing all old LoRA branches. In this way, they avoid forgetting caused by directly updating the LoRA parameters of old tasks. However, to handle the practical CL scenario

---

[*]Wu-Jun Li is the corresponding author.

39th Conference on Neural Information Processing Systems (NeurIPS 2025).

where task identities are unavailable at inference time, existing methods [32, 50, 61] based on LoRA integrate new and old LoRA branches through a simple addition. Consequently, they force the new and old LoRA branches to influence old tasks equally, which means that the new LoRA branch may cause a relatively large change in the model's output on old tasks. This leads to forgetting and degrades the model's overall performance in CL.

In this work, we propose a new method, called gated integration of low-rank adaptation (GainLoRA), for CL of LLMs. The contributions of GainLoRA are listed as follows:

- GainLoRA expands a new LoRA branch to learn each new task and introduces gating modules to integrate the new and old LoRA branches.

- GainLoRA leverages the new gating module to minimize the influence from the new LoRA branch to old tasks, effectively mitigating forgetting and improving the model's overall performance.

- Experimental results on CL benchmarks show that GainLoRA outperforms existing state-of-the-art CL methods.

## 2 Related Work and Preliminaries

### 2.1 Related Work

**Parameter-Efficient Fine-Tuning** Parameter-efficient fine-tuning (PEFT) methods tune a limited number of parameters to adapt a pre-trained model for downstream tasks, showing much higher efficiency than tuning all the parameters of the pre-trained model, especially for LLMs [72]. For example, Adapter [20] modifies the model architecture by introducing trainable modules into Transformer layers and tunes these modules for downstream tasks. Prompt-tuning [26] and Prefix-tuning [27] insert learnable tokens into the input and tune them for downstream tasks. Low-rank adaptation (LoRA) [21] reparameterizes the original model parameters with low-rank matrices and tunes these matrices for downstream tasks. Although tuning significantly fewer parameters than full fine-tuning, PEFT can achieve comparable performance to full fine-tuning across a wide range of natural language processing (NLP) tasks [16, 21, 37, 70].

**Continual Learning** There are three main types of CL methods, categorized as regularization-based methods, memory-based methods, and expansion-based methods. Regularization-based methods [23, 24, 29] incorporate a regularization term to mitigate catastrophic forgetting. Memory-based methods [10, 31, 34, 52, 75] utilize memory mechanisms to preserve knowledge from old tasks. Expansion-based methods [22, 28, 30, 46] mitigate catastrophic forgetting by introducing new parameters for learning new tasks while typically freezing old parameters.

Many CL methods [1, 29, 30] are designed to train models from scratch. Recent studies [32, 45, 56, 61, 67] have shown that leveraging PEFT strategies to fine-tune pre-trained models enables CL methods to achieve superior performance across tasks. For example, some methods [42, 45, 67, 75] use prompt-tuning for continual learning. They either maintain independent prompts for each task or maintain a prompt pool and perform query-key matching to learn new tasks. There are also many methods [32, 49, 61, 75] adopting LoRA for continual learning. Most of these methods expand a new LoRA branch to handle each new task while freezing old LoRA branches to mitigate catastrophic forgetting. However, they force the new and old LoRA branches to influence old tasks equally, potentially leading to forgetting.

### 2.2 Preliminaries

**Problem Definition** We follow existing CL works [61, 75] to formalize the problem definition for CL of LLMs. Specifically, in CL, a sequence of tasks $\{\mathcal{T}_1, \mathcal{T}_2, ..., \mathcal{T}_T\}$ is presented to the model sequentially, where $T$ denotes the total number of tasks. The $t$-th task $\mathcal{T}_t$ consists of a training dataset $\mathcal{D}_t$. For any given sample $(\boldsymbol{x}_t, \boldsymbol{y}_t) \in \mathcal{D}_t$, $\boldsymbol{x}_t$ denotes an input sentence and $\boldsymbol{y}_t$ denotes the corresponding output. When learning the $t$-th new task, the model is required to mitigate catastrophic forgetting of the $t - 1$ previously learned tasks.

Similar to existing CL works for LLMs [3, 75], we consider a more challenging CL setting with three key challenges: (1) the model is presented with a sequence of tasks spanning various types, such

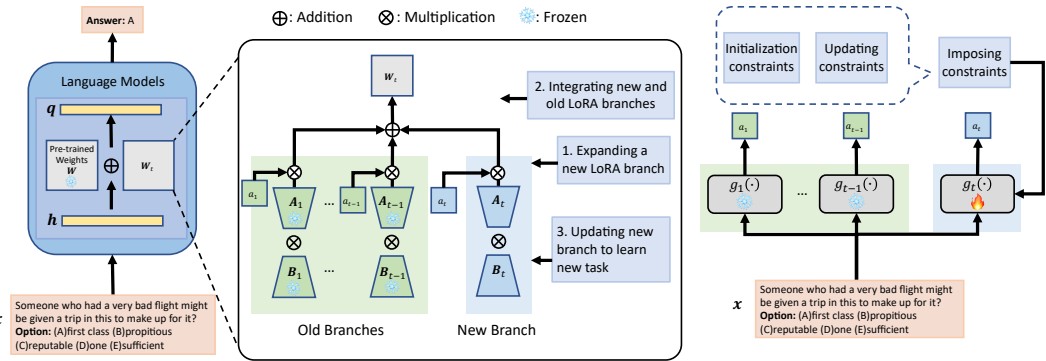

(a) Expandable LoRA Architecture in GainLoRA      (b) Gating Modules in GainLoRA

Figure 1: (a) shows the expandable LoRA architecture of our GainLoRA for learning the $t$-th new task. (b) shows that for each task $\mathcal{T}_i$, GainLoRA uses an independent gating module $g_i(\cdot)$ to generate integration coefficient $a_i$.

as dialogue generation, information extraction and so on; (2) the model is not provided with task identities at inference time; (3) the model must learn without access to real or synthetic samples from previously learned tasks.

**Low-Rank Adaptation** LoRA [21] is a widely adopted PEFT method used for fine-tuning various pre-trained models, particularly LLMs. Specifically, let $\boldsymbol{W} \in \mathbb{R}^{d_{out} \times d_{in}}$ represent a pre-trained weight in LLMs, where $d_{in}$ and $d_{out}$ are the input and output dimensions, respectively. LoRA introduces an additional branch consisting of two matrices, $\boldsymbol{A} \in \mathbb{R}^{d_{out} \times r}$ and $\boldsymbol{B} \in \mathbb{R}^{r \times d_{in}}$, where $r \ll \min(d_{in}, d_{out})$. LoRA then modifies the forward propagation of this layer as

$$\boldsymbol{e} = (\boldsymbol{W} + \boldsymbol{AB})\boldsymbol{h}.$$

Here, $\boldsymbol{h}$ and $\boldsymbol{e}$ denote the input and output, respectively. $\boldsymbol{A}$ is initialized to $\boldsymbol{0}$, and $\boldsymbol{B}$ is initialized with a Gaussian distribution. During fine-tuning for downstream tasks, the pre-trained weight $\boldsymbol{W}$ remains frozen, and only the parameters $\boldsymbol{A}$ and $\boldsymbol{B}$ are fine-tuned.

## 3 Methodology

Our GainLoRA employs an expandable LoRA architecture, which is illustrated in Figure 1 (a). Specifically, before learning the $t$-th task ($1 \leq t \leq T$), GainLoRA first expands the LoRA architecture by introducing the $t$-th new branch with matrices $\boldsymbol{A}_t \in \mathbb{R}^{d_{out} \times r}$ and $\boldsymbol{B}_t \in \mathbb{R}^{r \times d_{in}}$. The new and old LoRA branches are then integrated as

$$\boldsymbol{W}_t = \boldsymbol{W}_{t-1} + a_t \boldsymbol{A}_t \boldsymbol{B}_t = \sum_{i=1}^{t} a_i \boldsymbol{A}_i \boldsymbol{B}_i, \tag{1}$$

where $a_i$ is an integration coefficient that determines the influence of the $i$-th LoRA branch to the input $\boldsymbol{h}$. Note that $\boldsymbol{W}_{t-1}$ is a zero matrix when $t = 1$. As a result, the forward propagation in this layer is modified as

$$\boldsymbol{e} = (\boldsymbol{W} + \boldsymbol{W}_t)\boldsymbol{h}. \tag{2}$$

Finally, only the new LoRA branch (i.e. the $t$-th LoRA branch) is updated for the $t$-th new task, while all the old LoRA branches are frozen. After learning the $t$-th task, (2) is also used for inference across all test samples, thereby ensuring compatibility with the scenario where task identities are unavailable during inference.

Many existing CL methods based on LoRA [32, 49, 50, 61, 75] share a similar architecture to our method, as illustrated in Figure 1 (a). However, these methods fix all coefficients $\{a_i\}_{i=1}^{t}$ in (1) to 1, forcing the new and old LoRA branches influence old tasks equally. As a result, the new LoRA branch introduces a change of $\boldsymbol{A}_t \boldsymbol{B}_t \boldsymbol{h}$ to the output for inputs $\boldsymbol{h}$ associated with old tasks, potentially leading to forgetting. Although some methods attempt to mitigate this forgetting by

imposing regularization [50] or orthogonality constraints [32] on the new LoRA branch, the fixed integration coefficients $\{a_i\}_{i=1}^{t}$ still limit their performance, as demonstrated by the experimental results presented in Section 4. The method in [75] does not force the new and old LoRA branches to influence old tasks equally but relies on replaying synthetic old samples to mitigate forgetting, making it unsuitable for the scenario considered in this work.

Different from existing methods, GainLoRA introduces an independent gating module $g_i(\cdot)$ for each task $\mathcal{T}_i$ to generate the integration coefficients ($1 \leq i \leq T$). To mitigate the forgetting caused by the new task, GainLoRA leverages the gating module to minimize the influence from the new LoRA branch to the old tasks. The details will be introduced in the following subsections.

## 3.1 Architecture of Gating Modules

As illustrated in Figure 1 (b), given an input sample $\boldsymbol{x}$, the gating module $g_i(\cdot)$ generates the integration coefficient for the $i$-th LoRA branch, denoted as $a_i = g_i(\boldsymbol{x})$. The computation of $g_i(\cdot)$ is defined as

$$
\begin{aligned}
\boldsymbol{p}_{i,0} &= \boldsymbol{p}_0 = \mathrm{Pool}(\mathrm{Token}(\boldsymbol{x})), \\
\boldsymbol{p}_{i,l} &= \sigma(\boldsymbol{G}_{i,l}\boldsymbol{p}_{i,l-1}), \ l \in \{1, 2, ..., L\}, \\
g_i(\boldsymbol{x}) &= f(\boldsymbol{G}_{i,L+1}\boldsymbol{p}_{i,L}).
\end{aligned}
\tag{3}
$$

Here, $\mathrm{Token}(\cdot)$ represents the tokenizer used in LLMs to extract token embeddings from the input $\boldsymbol{x}$. $\mathrm{Pool}(\cdot)$ denotes an average pooling operation applied to the token embeddings to produce a fixed-size vector. $\sigma(\cdot)$ denotes the non-linear activation function. $\boldsymbol{G}_{i,l}$ denotes the weight matrix for the $l$-th layer of $g_i(\cdot)$ ($1 \leq l \leq L + 1$). In the final layer, $\boldsymbol{G}_{i,L+1}$ is a vector that maps the input vector $\boldsymbol{p}_{i,L}$ to a scalar. Following existing works with gating mechanisms [7, 19], the function $f(\cdot)$ is designed to map a scalar to a value within $[0, 1]$, that is, $f(\cdot) : \mathbb{R} \to [0, 1]$.

Note that the input to gating modules is the same as that of LLMs, denoted as $\boldsymbol{x}$, which differs from the input to LoRA in a specific layer, denoted as $\boldsymbol{h}$. During the learning of the $t$-th new task, only the new gating module $g_t(\cdot)$ is updated, while all the old gating modules $\{g_i(\cdot)\}_{i=1}^{t-1}$ remain frozen.

## 3.2 Minimizing the Influence from the New LoRA Branch to Old Tasks

GainLoRA minimizes the influence from the new LoRA branch to old tasks by making $a_t = g_t(\boldsymbol{x})$ as close to 0 as possible for any input $\boldsymbol{x}$ from old tasks $\{\mathcal{T}_i\}_{i=1}^{t-1}$. However, since we focus on the scenario where no real or synthetic samples from old tasks are accessible, directly optimizing $g_t(\boldsymbol{x})$ to 0 is impractical. To overcome this challenge, GainLoRA imposes constraints on the new gating module $g_t(\cdot)$, implicitly guiding $g_t(\boldsymbol{x})$ to close to 0 and reduce the influence of the new LoRA branch to old tasks.

In the following two subsections, we first describe the constraints imposed on the new gating module $g_t(\cdot)$ and explain how these constraints guide $g_t(\boldsymbol{x})$ close to 0 for any $\boldsymbol{x}$ from the old tasks. Then, we detail the implementation of these constraints during training.

### 3.2.1 Constraints on New Gating Module

To formalize the constraints imposed on the new gating module $g_t(\cdot)$, we define the subspace spanned by the inputs to $\boldsymbol{G}_{t,l}$ ($1 \leq l \leq L + 1$) from the previous $t - 1$ tasks as:

$$
\mathcal{M}_{t,l} = \mathrm{span}\{\boldsymbol{p}_{t,l-1} | \ \boldsymbol{p}_{t,l-1} \text{ is defined in (3)}, (\boldsymbol{x}, \boldsymbol{y}) \in \cup_{i=1}^{t-1}\mathcal{D}_i\}.
\tag{4}
$$

Note that subspaces $\{\mathcal{M}_{t,l}\}_{l=1}^{L+1}$ cannot be obtained directly due to the unavailability of samples from old tasks. However, by introducing additional constraints, $\{\mathcal{M}_{t,l}\}_{l=1}^{L+1}$ can be solved iteratively, which will be discussed in Section 3.2.2.

**Initialization Constraints** Before learning the $t$-th task, the following constraints are imposed on the initialization of the new gating module $g_t(\cdot)$:

$$
\mathrm{Init}(\boldsymbol{G}_{t,L+1}) \perp \mathcal{M}_{t,L+1}, \ f(0) = 0,
\tag{5}
$$

where $\mathrm{Init}(\boldsymbol{G}_{t,L+1})$ denotes the initialization of $\boldsymbol{G}_{t,L+1}$. These constraints ensure that for any sample $\boldsymbol{x}$ from the old tasks, the integration coefficient satisfies

$$
a_t = g_t(\boldsymbol{x}) = f(\mathrm{Init}(\boldsymbol{G}_{t,L+1})\boldsymbol{p}_{t,L}) = 0,
\tag{6}
$$

where $\boldsymbol{p}_{t,L}$ is defined in (3). The second equality holds since $\boldsymbol{G}_{t,L+1} = \text{Init}(\boldsymbol{G}_{t,L+1})$ before learning the $t$-th new task. The third equality holds because $f(0) = 0$ and $\boldsymbol{p}_{t,L} \in \mathcal{M}_{t,L+1}$ for any $\boldsymbol{x}$ from previous $t-1$ tasks.

**Updating Constraints** During the learning of the $t$-th task, the following constraints are imposed on the updates to the new gating module $g_t(\cdot)$:

$$\Delta\boldsymbol{G}_{t,l} \perp \mathcal{M}_{t,l} \quad \text{for} \quad 1 \leq l \leq L+1, \tag{7}$$

where $\Delta\boldsymbol{G}_{t,l}$ denotes the update to $\boldsymbol{G}_{t,l}$. Based on existing studies [60, 30], the constraints in (7) ensure that $g_t(\boldsymbol{x})$ remains unchanged for inputs $\boldsymbol{x}$ from the old tasks during the learning of the $t$-th task. Formally, the following proposition holds:

**Proposition 3.1.** *If the constraints in (7) are satisfied, subspaces $\{\mathcal{M}_{t,l}\}_{l=1}^{L+1}$ remain unchanged during the learning of the $t$-th task. Furthermore, for any input $\boldsymbol{x}$ from the previous $t-1$ tasks, $g_t(\boldsymbol{x})$ remains unchanged during the learning of the $t$-th task.*

The proof of this proposition is provided in Appendix A.3. Since the initialization constraints in (5) ensure $g_t(\boldsymbol{x}) = 0$ before learning the $t$-th new task, $g_t(\boldsymbol{x}) = 0$ is preserved throughout the learning process if the updating constraints in (7) are satisfied.

The fact that subspaces $\{\mathcal{M}_{t,l}\}_{l=1}^{L+1}$ remain unchanged, as stated in Proposition 3.1, is essential for implementing the orthogonal constraints in (7). Specifically, as will be detailed in Section 3.2.2, orthonormal bases for the subspaces $\{\mathcal{M}_{t,l}\}_{l=1}^{L+1}$ are learned to enforce the orthogonal constraints in (5) and (7). Since the subspaces $\{\mathcal{M}_{t,l}\}_{l=1}^{L+1}$ remain unchanged during the learning of the $t$-th task, their orthonormal bases also remain unchanged, allowing them to be pre-computed before learning the $t$-th task, thus facilitating the implementation of orthogonal constraints in (5) and (7) throughout the learning process.

### 3.2.2 Implementation of Constraints

There exist many functions $f(\cdot) : \mathbb{R} \to [0, 1]$ satisfying $f(0) = 0$. In this work, we define $f(\cdot)$ as

$$f(b) = |2 \cdot \text{sigmoid}(b) - 1|, \tag{8}$$

where $\text{sigmoid}(\cdot)$ denotes the sigmoid function. Other functions $f(\cdot) : \mathbb{R} \to [0, 1]$ that satisfy $f(0) = 0$ are also applicable, and experiments with different choices of $f(\cdot)$ are provided in Appendix C.3.1. Better model performance can be expected by designing more effective $f(\cdot)$, but this is not the focus of this paper.

Implementing the orthogonal constraints in (5) and (7) is challenging due to the lack of samples from previous $t-1$ tasks to approximate the subspaces $\{\mathcal{M}_{t,l}\}_{l=1}^{L+1}$. To address this issue, we further impose the following constraints on the initialization of $\boldsymbol{G}_{t,l}$ ($1 \leq l \leq L$):

$$\text{Init}(\boldsymbol{G}_{t,l}) \leftarrow \boldsymbol{G}_{t-1,l}. \tag{9}$$

This strategy initializes the first $L$ layers of $g_t(\cdot)$ using the corresponding layers from the previous gating module $g_{t-1}(\cdot)$. As a result, the first $L$ layers of $g_t(\cdot)$ can be viewed as being initialized and starting their training at the beginning of the first task, continuing until the $t$-th task. Simultaneously, the first $L$ layers in $g_i(\cdot)$ serve as checkpoints, preserving the state of $g_t(\cdot)$ after learning the $i$-th task ($1 \leq i \leq t$). At this time, we can use existing method gradient projection memory (GPM) [47] to iteratively learn a set of matrices $\{\boldsymbol{M}_{t,l}\}_{l=1}^{L+1}$, where the columns of $\boldsymbol{M}_{t,l}$ contribute to a set of orthonormal bases of subspace $\mathcal{M}_{t,l}$. Details of GPM are provided in Appendix A.1. Then, before learning the $t$-th task, the following operation can be performed on $\text{Init}(\boldsymbol{G}_{t,L+1})$:

$$\text{Init}(\boldsymbol{G}_{t,L+1}) \leftarrow \text{Init}(\boldsymbol{G}_{t,L+1}) - \boldsymbol{M}_{t,L+1}\boldsymbol{M}_{t,L+1}^T\text{Init}(\boldsymbol{G}_{t,L+1}). \tag{10}$$

At this time, we have

$$\begin{aligned}
&\boldsymbol{M}_{t,L+1}^T(\text{Init}(\boldsymbol{G}_{t,L+1}) - \boldsymbol{M}_{t,L+1}\boldsymbol{M}_{t,L+1}^T\text{Init}(\boldsymbol{G}_{t,L+1})) \\
=&\boldsymbol{M}_{t,L+1}^T(\boldsymbol{I} - \boldsymbol{M}_{t,L+1}\boldsymbol{M}_{t,L+1}^T)\text{Init}(\boldsymbol{G}_{t,L+1}) \\
=&(\boldsymbol{I} - \boldsymbol{M}_{t,L+1}^T\boldsymbol{M}_{t,L+1})\boldsymbol{M}_{t,L+1}^T\text{Init}(\boldsymbol{G}_{t,L+1}).
\end{aligned} \tag{11}$$

Since the columns of $\boldsymbol{M}_{t,L+1}$ form an orthonormal basis, we have $\boldsymbol{M}_{t,L+1}^T\boldsymbol{M}_{t,L+1} = \boldsymbol{I}$, which means $\boldsymbol{I} - \boldsymbol{M}_{t,L+1}^T\boldsymbol{M}_{t,L+1} = \boldsymbol{O}$. Therefore, the equation in (11) is equal to zero matrix $\boldsymbol{O}$. Note

---

**Algorithm 1** GainLoRA for Continual Learning

---
**Input:** The data of different tasks $\{\mathcal{D}_t\}_{t=1}^T$.
**Output:** Learned LoRA parameters $\{(\boldsymbol{A}_i, \boldsymbol{B}_i)\}_{i=1}^T$ and gating modules $\{g_i(\cdot)\}_{i=1}^T$.
**for** $t$ in $1 : T$ **do**
    Expand the $t$-th new LoRA branch with $\boldsymbol{A}_t$ and $\boldsymbol{B}_t$;
    Impose initialization constraints on the new gating module $g_t(\cdot)$ by (8), (9) and (10);
    Integrate new and old LoRA branches by (1);
    **for** $\mathcal{B}_t \subseteq \mathcal{D}_t$ **do**
        Compute the loss in (13);
        Perform backward propagation to compute the update of the new LoRA branch and the new gating module;
        Impose updating constraints on the update of the new gating module by (7);
    **end for**
**end for**

---

that $\mathcal{M}_{t,L+1}$ is spanned by the columns of $\boldsymbol{M}_{t,L+1}$, $\mathrm{Init}(\boldsymbol{G}_{t,L+1})$ satisfies the constraints in (5) after the operation in (10).

Similarly, during the learning of the $t$-th task, the following operation can be performed on $\{\Delta \boldsymbol{G}_{t,l}\}_{l=1}^{L+1}$:

$$\Delta \boldsymbol{G}_{t,l} \leftarrow \Delta \boldsymbol{G}_{t,l} - \boldsymbol{M}_{t,l}\boldsymbol{M}_{t,l}^T \Delta \boldsymbol{G}_{t,l}. \tag{12}$$

With the same proving process in (11), we can show that the update in (12) allows the update $\{\Delta \boldsymbol{G}_{t,l}\}_{l=1}^{L+1}$ to satisfy the constraints in (7).

### 3.3 Updating the New LoRA Branch

Our GainLoRA aims to effectively integrate new and old LoRA branches while mitigating forgetting caused by the new LoRA branch on old tasks. Since GainLoRA does not impose specific update strategies for the new LoRA branch, it is inherently compatible with various existing CL methods that adopt similar LoRA architecture as our method and can update the new LoRA branch [32, 50, 61]. Since these existing methods fix all integration coefficients $\{a_i\}_{i=1}^t$ to 1, combining our method with these existing methods can enhance their performance, as demonstrated in Section 4.

### 3.4 Whole Process of GainLoRA

Algorithm 1 outlines the whole process of our GainLoRA. Before learning the $t$-th new task $\mathcal{T}_t$, GainLoRA first expands the LoRA architecture by introducing the $t$-th new branch with matrices $\boldsymbol{A}_t$ and $\boldsymbol{B}_t$. Simultaneously, a new gating module $g_t(\cdot)$ is initialized through the operations specified in (8), (10) and (9) to ensure that the initialization constraints in (5) are satisfied. The new and old LoRA branches are then integrated using (1), and the forward propagation is modified as (2).

During the learning of the $t$-th task $\mathcal{T}_t$ with the corresponding dataset $\mathcal{D}_t$, our method follows existing methods [61, 75] and computes the loss for the new task through

$$\mathcal{L}_t = \frac{1}{|\mathcal{D}_t|} \sum_{(\boldsymbol{x}_t, \boldsymbol{y}_t) \in \mathcal{D}_t} \sum_{j=1}^{|\boldsymbol{y}_t|} \log\left[P(y_{t,j}|\boldsymbol{x}_t, y_{t,1}, ..., y_{t,j-1})\right], \tag{13}$$

where $\boldsymbol{y}_t = [y_{t,1}, y_{t,2}, ..., y_{t,|\boldsymbol{y}_t|}]$. Each time, GainLoRA samples a mini-batch $\mathcal{B}_t$ to minimize the loss in (13) by updating the new LoRA branch and the new gating module $g_t(\cdot)$. During this process, the projections defined in (12) are applied to the parameters of $g_t(\cdot)$, ensuring that the update constraints in (7) are satisfied.

GainLoRA introduces a new gating module for each new task, which incurs additional parameters and computational cost when combined with other methods. Section 4 will demonstrate that the trainable parameters added by GainLoRA are limited, making the number of trainable parameters in GainLoRA comparable to other methods. Additionally, Appendix C.1 and C.2 will demonstrate that the computational cost introduced by GainLoRA is minimal compared to the original LLMs.

## 4 Experiments

### 4.1 Experimental Settings

**Datasets** Following existing CL methods [45, 61, 75], we evaluate different methods on SuperNI [64] and Long Sequence [45] benchmarks. SuperNI benchmark includes various types of NLP tasks, including dialogue generation, information extraction, question answering, summarization, and sentiment analysis. Following the protocols of existing method [75], three tasks are selected from each type, resulting in 15 tasks. These tasks are arranged into two different task sequences with different orders, referred to as Order 1 and Order 2. Long Sequence benchmark consists of 15 diverse classification tasks, which are similarly arranged into two task sequences with different orders, referred to as Order 3 and Order 4. More details about the benchmarks and task sequences are provided in Appendix B.

**Evaluation Metric** We use $A_{j,i}$ to denote the model's performance on the $i$-th task once the model learns the $j$-th task. Specifically, $A_{j,i}$ represents accuracy for classification tasks and Rouge-L [33] for other types of tasks. Following traditional CL works [5, 6], we employ average performance (AP) and forgetting (FT) to evaluate the model's performance. The formulas for these two metrics are defined as

$$\text{AP} = \frac{1}{T} \sum_{i=1}^{T} A_{T,i}, \quad \text{FT} = \frac{1}{T-1} \sum_{i=1}^{T-1} (\max_{l \in \{1,2,\dots,T-1\}} A_{l,i} - A_{T,i}), \tag{14}$$

where $T$ denotes the total number of tasks in the task sequence. AP evaluates the model's final performance, and FT quantifies the forgetting.

**Baselines** We compare our method with state-of-the-art CL methods, including LFPT5 [42], EPI [65], MIGU [11], EWC [24], TASL [15], KIFLoRA [14], IncLoRA [61], C-LoRA [50], O-LoRA [61], and InfLoRA [32]. Additionally, we introduce a simple baseline called SeqLoRA, which does not expand new LoRA branches but sequentially updates old LoRA parameters for new tasks and lacks mechanism to mitigate forgetting. Note that many CL methods based on pre-trained models in CV focus on classification tasks, relying either on carefully designed classifiers [38, 71, 74] or the [CLS] token in ViT [25, 51, 57, 58, 63, 66, 67]. In contrast, we follow existing works in NLP [61, 75] and adopt next-token prediction to handle both classification and generation tasks, where models [43, 54] lack a [CLS] token. Consequently, these CV methods are incompatible with our setting and cannot be directly compared. For completeness, we adapt some of them to our setup and report results in Appendix C.7.

**Implementation Details** Following existing CL works [40, 61, 68], all methods are implemented with instruction tuning [40] and optimized using AdamW [35]. To ensure fair comparisons, for all the methods based on LoRA, we follow existing CL methods [21, 61, 75] by incorporating the LoRA architecture into the query and value components of the multi-head attention mechanism in each Transformer block. Similar to the existing CL methods for LLMs [61, 75], we use T5 [43], Llama-2 [54] and Llama-3 [12] as the base architectures. Each experiment is repeated three times with different seeds, and the average result is reported. More details, such as the learning rate, batch size, and architecture of the gating modules in GainLoRA, are provided in Appendix B.2 and Appendix B.3.

### 4.2 Experimental Results

**Compare with Existing Methods** We first follow existing works [11, 75] and evaluate different CL methods using T5-Large. Since our method does not impose specific update strategies for the new LoRA branch, we adopt the same update strategies as the two state-of-the-art methods, O-LoRA [61] and InfLoRA [32]. Note that these two methods leverage LoRA architecture similar to our method but fix all integration coefficients $\{a_i\}_{i=1}^{T}$ to 1. Details of these two methods are provided in Appendix A.2. We use GainLoRA (O-LoRA) and GainLoRA (InfLoRA) to respectively denote our methods adopting O-LoRA and InfLoRA to update the new LoRA branch. GainLoRA is also compatible with other methods that leverage expandable LoRA architecture shown in Figure 1 (a), and we give some results in Appendix C.5.

The results are shown in Table 1. As we can see, our methods GainLoRA (O-LoRA) and Gain-LoRA (InfLoRA) outperform O-LoRA and InfLoRA in both AP and FT, respectively. This im-

Table 1: Results on different task sequences with T5-large model. Results of methods with * are copied from existing paper [75].

| Method | Order 1 | | Order 2 | | Order 3 | | Order 4 | |
| --- | --- | --- | --- | --- | --- | --- | --- | --- |
| | AP↑ | FT↓ | AP↑ | FT↓ | AP↑ | FT↓ | AP↑ | FT↓ |
| LFPT5* [42] | 39.03 | 10.87 | 29.70 | 20.72 | 66.62 | 14.57 | 67.40 | 13.20 |
| EPI* [65] | - | - | - | - | 75.19 | 0.77 | 75.10 | 2.44 |
| MIGU+FT [11] | - | - | - | - | 71.30 | 11.39 | 69.05 | 14.06 |
| EWC [24] | 15.32 | 26.78 | 18.19 | 30.28 | 43.24 | 23.66 | 46.25 | 32.90 |
| TaSL [15] | 27.51 | 18.53 | 28.05 | 17.39 | 71.37 | 6.20 | 73.11 | 6.52 |
| KIFLoRA [14] | 28.33 | 16.44 | 30.31 | 16.27 | 72.19 | 3.10 | 73.72 | 4.75 |
| SeqLoRA | 7.30 | 47.60 | 7.03 | 47.97 | 49.46 | 27.60 | 33.81 | 45.53 |
| IncLoRA [61] | 12.33 | 41.93 | 16.65 | 36.56 | 61.19 | 13.63 | 62.46 | 15.92 |
| C-LoRA [50] | 22.69 | 24.25 | 32.81 | 11.60 | 66.83 | 8.64 | 61.86 | 14.18 |
| O-LoRA [61] | 26.37 | 19.15 | 32.83 | 11.99 | 70.98 | 3.69 | 71.21 | 4.03 |
| GainLoRA (O-LoRA) | **47.84** | **2.26** | **46.84** | 2.91 | 73.37 | 3.02 | 76.01 | 2.49 |
| InfLoRA [32] | 39.78 | 7.64 | 39.57 | 8.93 | 75.15 | 4.19 | 75.79 | 3.47 |
| GainLoRA (InfLoRA) | 46.21 | 2.40 | 46.44 | **2.61** | **78.01** | **0.77** | **77.54** | **1.25** |

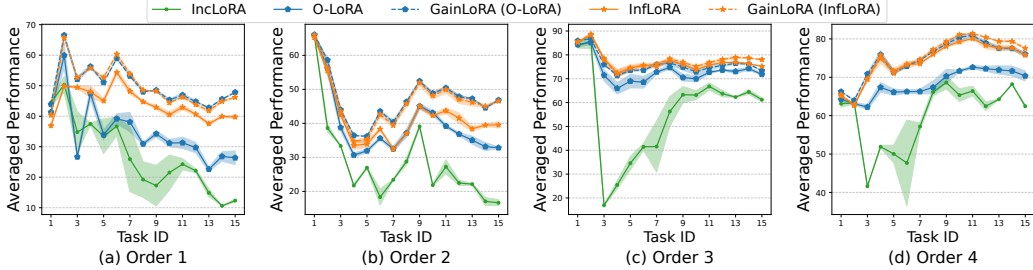

Figure 2: The variation of performance across different CL methods during training on different task sequences.

provement demonstrates that fixing all coefficients $\{a_i\}_{i=1}^{T}$ to 1 leads to forgetting on old tasks, thereby limiting the performance of O-LoRA and InfLoRA. By effectively mitigating this forgetting, GainLoRA (O-LoRA) and GainLoRA (InfLoRA) achieve superior performance. Furthermore, our methods consistently achieve the best performance across all task sequences.

Figure 2 illustrates the variation in the average performance across all learned tasks for different methods throughout the CL process. As shown, GainLoRA consistently outperforms the performance of O-LoRA and InfLoRA throughout the whole training process.

**Scaling to Larger Model Architectures** To evaluate the effectiveness of our method on larger model architectures, we scale different LoRA-based CL methods to larger models, including T5-XL, Llama-2-7B, Llama-2-13B and Llama-3-8B. Table 2 and Table 3 present the results of different methods. As shown, across models of varying sizes, GainLoRA (O-LoRA) and GainLoRA (InfLoRA) consistently outperform O-LoRA and InfLoRA in terms of AP and FT, respectively. This demonstrates that GainLoRA effectively mitigates forgetting in the new LoRA branch across different model architectures.

**Trainable Parameters** We compare the number of trainable parameters across different methods for training on different task sequences. The results for T5, Llama-2 and Llama-3 are shown in Figure 3, and the detailed computation of trainable parameters is provided in Appendix B.4. As shown, GainLoRA (O-LoRA) and GainLoRA (InfLoRA) have more trainable parameters than O-LoRA and InfLoRA, respectively. This increase arises from the introduction of the trainable gating module in GainLoRA. However, the additional trainable parameters introduced by GainLoRA are much fewer than those in LoRA. Therefore, the total number of trainable parameters in GainLoRA (O-LoRA) and GainLoRA (InfLoRA) are comparable to that of O-LoRA and InfLoRA, respectively.

**Distribution of Outputs in New Gating Module** To demonstrate that our GainLoRA effectively minimizes the influence from the new LoRA branches to old tasks, we analyze the output distributions

Table 2: The overall results on different task sequences with T5-XL model.

| Method | Order 1 AP↑ | Order 1 FT↓ | Order 2 AP↑ | Order 2 FT↓ | Order 3 AP↑ | Order 3 FT↓ | Order 4 AP↑ | Order 4 FT↓ |
|---|---|---|---|---|---|---|---|---|
| O-LoRA [61] | 36.50 | 11.42 | 40.64 | 6.37 | 73.77 | 2.70 | 76.19 | 3.56 |
| GainLoRA (O-LoRA) | **50.10** | 3.21 | 49.86 | 3.04 | 78.41 | 2.59 | 77.21 | 3.30 |
| InfLoRA [32] | 45.61 | 5.60 | 45.85 | 5.10 | 80.22 | 2.09 | 79.43 | 1.71 |
| GainLoRA (InfLoRA) | 50.06 | **1.86** | **50.26** | **2.64** | **81.22** | **0.58** | **80.30** | **0.75** |

Table 3: The overall results on different task sequences with Llama-2-7B, Llama-2-13B and Llama-3-8B.

| Models | Methods | Order 1 AP↑ | Order 1 FT↓ | Order 2 AP↑ | Order 2 FT↓ |
|---|---|---|---|---|---|
| Llama-2-7B | O-LoRA [61] | 39.37 | 15.84 | 37.55 | 20.23 |
| | GainLoRA (O-LoRA) | 51.10 | 4.96 | **51.14** | 5.57 |
| | InfLoRA [32] | 42.93 | 11.23 | 39.94 | 15.00 |
| | GainLoRA (InfLoRA) | **51.27** | **2.84** | 50.17 | **4.71** |
| Llama-2-13B | O-LoRA [61] | 43.92 | 14.15 | 40.05 | 19.53 |
| | GainLoRA (O-LoRA) | 52.47 | 4.78 | 51.68 | 5.86 |
| | InfLoRA [32] | 43.64 | 14.85 | 45.74 | 10.61 |
| | GainLoRA (InfLoRA) | **53.64** | **2.87** | **52.46** | **4.90** |
| Llama-3-8B | O-LoRA [61] | 42.49 | 8.85 | 38.67 | 19.28 |
| | GainLoRA (O-LoRA) | **53.39** | 3.56 | 51.69 | 6.20 |
| | InfLoRA [32] | 43.27 | 6.02 | 48.77 | 5.88 |
| | GainLoRA (InfLoRA) | 52.18 | **1.40** | **52.48** | **4.21** |

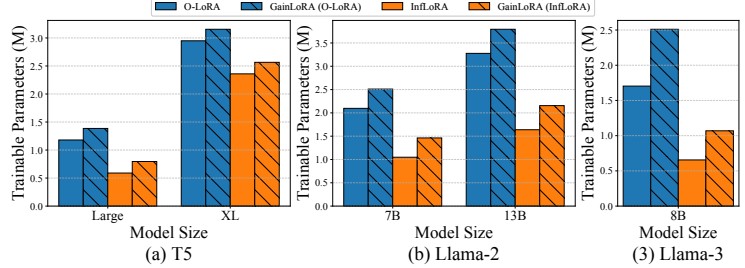

Figure 3: (a), (b) and (c) show the number of trainable parameters for different CL methods and model backbones under task sequences Order 1 and Order 2.

of the new gating modules. Specifically, after training on the final task (i.e., the 15-th task) in the task sequences, the 15-th task corresponds to the new task, and its associated gating module $g_{15}(\cdot)$ serves as the new gating module.

We obtain the outputs of the new gating module $g_{15}(\cdot)$ on the samples from old and new tasks, respectively. Then, we analyze their distributions in Figure 4. As shown, the outputs of $g_{15}(\cdot)$ for the samples from old tasks are concentrated around 0, effectively minimizing the influence from the new LoRA branch to old tasks. Furthermore, GainLoRA does not constrain the outputs of $g_{15}(\cdot)$ for the samples from the new task. As a result, the outputs of $g_{15}(\cdot)$ for the samples from the new task are distributed near 1, enabling the model to effectively learn the new task.

**Ablation Study** To verify the necessity of both the initialization and updating constraints introduced in Section 3.2.1, we define several variants of GainLoRA. The first variant, referred to as "No Initialization Constraints", removes the initialization constraints defined in (5). Specifically, it replaces $f(\cdot)$ defined in (8) with function $\mathrm{sigmoid}(\cdot)$ and eliminates the operation in (10) while keeping all other components unchanged. The second variant, referred to as "No Updating Constraints", removes the updating constraints defined in (7) by eliminating the operations in (12) while preserving all other components of GainLoRA. The third variant, referred to as "No Constraints", follows "No

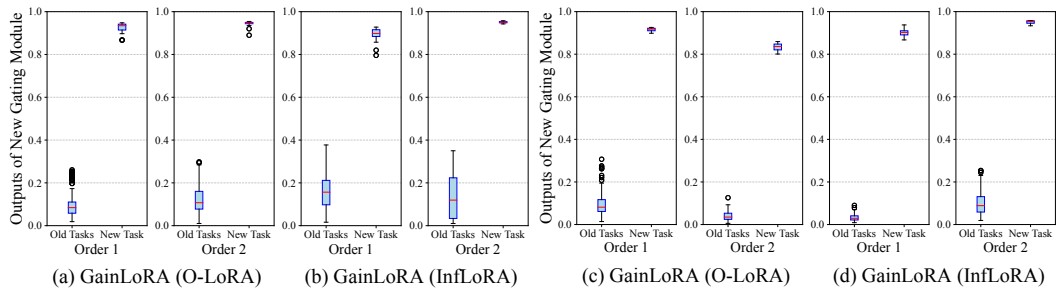

Figure 4: (a) and (b) show outputs of new gating module in our GainLoRA on different task sequences with T5-Large. (c) and (d) show outputs of new gating module in our GainLoRA on different task sequences with Llama-2-7B.

Table 4: Ablation study of GainLoRA with T5-Large and Llama-2-7B.

| Method | T5-Large | | | | Llama-2-7B | | | |
| | Order 1 | | Order 2 | | Order 1 | | Order 2 | |
| | AP↑ | FT↓ | AP↑ | FT↓ | AP↑ | FT↓ | AP↑ | FT↓ |
|---|---|---|---|---|---|---|---|---|
| GainLoRA (O-LoRA) | **47.84** | **2.26** | **46.84** | **2.91** | **51.10** | **4.96** | **51.14** | **5.57** |
| No Initialization Constraints | 35.30 | 17.19 | 39.82 | 12.90 | 44.02 | 11.71 | 42.89 | 14.77 |
| No Updating Constraints | 23.01 | 30.32 | 24.96 | 28.14 | 33.74 | 23.06 | 34.71 | 22.36 |
| No Constraints | 26.32 | 26.00 | 30.63 | 22.37 | 34.48 | 23.46 | 36.87 | 21.24 |
| GainLoRA (InfLoRA) | **46.21** | **2.40** | **46.44** | **2.61** | **51.27** | **2.84** | **50.17** | **4.71** |
| No Initialization Constraints | 45.38 | 3.40 | 43.05 | 5.15 | 50.48 | 3.48 | 48.17 | 6.45 |
| No Updating Constraints | 37.69 | 10.94 | 38.85 | 9.31 | 48.52 | 5.68 | 47.85 | 7.00 |
| No Constraints | 36.75 | 12.18 | 41.00 | 6.66 | 49.10 | 6.07 | 45.77 | 8.70 |

Initialization Constraints" and "No Updating Constraints" to remove both the initialization and updating constraints. Table 4 presents the experimental results of these variants. As shown, none of these variants perform as well as our GainLoRA, indicating the critical role of both the initialization constraints and updating constraints in our GainLoRA.

## 5 Conclusion

In this work, we propose a new method, called GainLoRA, for CL of language models. GainLoRA expands a new LoRA branch for each new task and introduces gating modules to integrate the new and old LoRA branches. Furthermore, GainLoRA leverages the new gating module to minimize the influence of the new LoRA branch to old tasks, effectively mitigating forgetting and improving the model's overall performance. Experimental results on CL benchmarks demonstrate that GainLoRA outperforms existing state-of-the-art methods.

**Limitations** Similar to many CL methods for LLMs [15, 61], our method imposes some constraints on the model to mitigate forgetting. While effective, these constraints may accumulate with increasing tasks, potentially hindering the learning of new tasks. Furthermore, consistent with existing works [24, 32, 61], our method primarily targets at catastrophic forgetting with non-overlapping tasks, and further investigation is needed to assess its effect on more complex scenarios, such as scenarios where there is overlap between tasks [39, 2].

## 6 Broader Impacts

Continual learning (CL) offers a promising direction for improving the efficiency and scalability of language models, particularly in settings with continuously arriving tasks. By enabling incremental updates without retraining from scratch, it significantly reduces computational overhead and resource demands. However, CL often introduces additional components (e.g. memory or gating mechanisms), increasing complexity and requiring effort for maintenance or deployment.

## Acknowledgment

This work is supported by NSFC (No.62192783).

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

# A  More Details of Methods

## A.1  Gradient Projection Memory

We initialize the first $L$ layers of $g_t(\cdot)$ using the corresponding layers from the previous gating module $g_{t-1}(\cdot)$. Therefore, the first $L$ layers of $g_t(\cdot)$ can be viewed as being initialized at the beginning of the first task and continue their training until the $t$-th task. Additionally, the first $L$ layers in $g_i(\cdot)$ serve as checkpoints, preserving the state of $g_t(\cdot)$ after learning the $i$-th task ($1 \leq i \leq t$). At this time, existing method gradient projection memory (GPM) [47] can be used to learn matrices $\{M_{t,l}\}_{l=1}^{L+1}$, where the columns of $M_{t,l}$ approximate the orthonormal bases of the subspace $\mathcal{M}_{t,l}$. Specifically, when $t = 1$, since there is no old task, $\mathcal{M}_{1,l}$ is a null space and $M_{1,l}$ is a zero matrix. After learning the $t$-th new task, GPM expands $\mathcal{M}_{t,l}$ to $\mathcal{M}_{t+1,l}$ by first computing the input matrix $H_{t,l}$ where each column of $H_{t,l}$ represents an input to the $l$-th layer. Then, the component of $H_{t,l}$ already in $\mathcal{M}_{t,l}$ is removed by

$$\widehat{H}_{t,l} = H_{t,l} - M_{t,l}(M_{t,l})^T H_{t,l}. \tag{15}$$

Next, singular value decomposition (SVD) is performed on $\widehat{H}_{t,l}\widehat{H}_{t,l}^T$, which is decomposed as $\widehat{U}_{t,l}\widehat{\Sigma}_{t,l}\widehat{U}_{t,l}^T$. Then, $u$ new orthonormal bases $u_1, ..., u_u$ are chosen from the columns of $\widehat{U}_{t,l}$, where $u$ is the minimum number satisfying the following criteria for a given threshold $\epsilon_{th}$:

$$||(\widehat{H}_{t,l})_u||_F^2 + ||M_{t,l}(M_{t,l})^T H_{t,l}||_F^2 \geq \epsilon_{th}||H_{t,l}||_F^2. \tag{16}$$

Here, $(\widehat{H}_{t,l})_u$ denotes the components of $\widehat{H}_{t,l}$ corresponding to the top-$u$ singular values. Then, the orthonormal bases of subspace $\mathcal{M}_{t+1,l}$ are obtained by augmenting the orthonormal bases of subspace $\mathcal{M}_{t,l}$ with the new orthogonal vectors $u_1, ..., u_u$, resulting in $M_{t+1,l} = [M_{t,l}, u_1, ..., u_u]$.

## A.2  More Details of O-LoRA and InfLoRA

**O-LoRA**  O-LoRA [61] ensures that the new LoRA branch remains orthogonal to all the old LoRA branches. Specifically, during the learning of the $t$-th new task with the $t$-th LoRA branch ($A_t, B_t$), O-LoRA computes the inner product between the new and old LoRA branches as

$$O_{i,t} = B_i^T B_t \quad \text{for } 1 \leq i \leq t - 1 \tag{17}$$

Then, the loss function of O-LoRA is defined as

$$\frac{1}{|\mathcal{D}_t|} \sum_{(x_t, y_t) \in \mathcal{D}_t} \sum_{j=1}^{|y_t|} \log\left[P(y_{t,j}|x_t, y_{t,1}, ..., y_{t,j-1})\right] + \lambda \sum_{i=1}^{t-1} \sum_{j,k} ||O_{i,t}[j,k]||_2^2 \tag{18}$$

For further details on O-LoRA, we refer readers to the original paper [61].

**InfLoRA**  InfLoRA [32] ensures orthogonality between the new LoRA branch and the gradients of old tasks. Specifically, it shows that only fine-tuning the down-projection matrix $A_t$ in the new LoRA branch is equivalent to directly fine-tuning the pre-trained weights within a subspace spanned by the rows of $B_t$. Therefore, before learning the $t$-th task, InfLoRA designs $B_t$ to be orthogonal to the gradients of the old tasks. During the learning of the $t$-th task, InfLoRA only tunes $A_t$ in the new LoRA branch while freezing $B_t$ and all the old LoRA branches. For further details on InfLoRA, we refer readers to the original paper [32].

## A.3  Proof of Proposition 3.1

**Proposition A.1.** *If the constraints in (7) are satisfied, subspaces $\{\mathcal{M}_{t,l}\}_{l=1}^{L+1}$ remain unchanged during the learning of the $t$-th task. Furthermore, for any input $x$ from the previous $t - 1$ tasks, $g_t(x)$ remains unchanged during the learning of the $t$-th task.*

*Proof.* For any $x$ from previous $t - 1$ tasks, we rewrite $g_t(x)$ as

$$\begin{aligned}
g_t(x) &= f(G_{t,L+1}p_{t,L}), \\
p_{t,l} &= \sigma(G_{t,l}p_{t,l-1}), \ l \in \{1, 2, ..., L\}, \\
p_{t,0} &= p_0 = \text{Pool}(\text{Token}(x)).
\end{aligned} \tag{19}$$

Since $\boldsymbol{p}_{t,0} = \mathrm{Pool}(\mathrm{Token}(\boldsymbol{x}))$ is unrelated to the parameters of the new gating module $g_t(\cdot)$, $\boldsymbol{p}_{t,0}$ does not change with the update of $g_t(\cdot)$. Since $\mathcal{M}_{t,1}$ is spanned by $\boldsymbol{p}_{t,0}$, $\mathcal{M}_{t,1}$ remains unchanged during the learning of the $t$-th task.

Suppose that we have proven that $\boldsymbol{p}_{t,l-1}$ does not change with the update of the new gating module $g_t(\cdot)$ ($1 \leq l \leq L$). Since $\mathcal{M}_{t,l}$ is spanned by $\boldsymbol{p}_{t,l-1}$, $\mathcal{M}_{t,l}$ remains unchanged during the learning of the $t$-th task. At this point, $\boldsymbol{p}_{t,l}$ can be expressed as

$$\boldsymbol{p}_{t,l} = \sigma((\mathrm{Init}(\boldsymbol{G}_{t,l}) + \Delta\boldsymbol{G}_{t,l})\boldsymbol{p}_{t,l-1}) = \sigma(\mathrm{Init}(\boldsymbol{G}_{t,l})\boldsymbol{p}_{t,l-1}). \tag{20}$$

Here, the second equality holds since $\boldsymbol{p}_{t,l-1} \in \mathcal{M}_{t,l}$ and $\Delta\boldsymbol{G}_{t,l}\perp\mathcal{M}_{t,l}$. Therefore, $\boldsymbol{p}_{t,l}$ does not change with the update of the new gating module $g_t(\cdot)$ ($1 \leq l \leq L$). Since $\mathcal{M}_{t,l+1}$ is spanned by $\boldsymbol{p}_{t,l}$, $\mathcal{M}_{t,l+1}$ remains unchanged during the learning of the $t$-th task.

Furthermore, during the learning of the $t$-th task, $g_t(\boldsymbol{x})$ can be expressed as

$$\boldsymbol{g}_t(\boldsymbol{x}) = f((\mathrm{Init}(\boldsymbol{G}_{t,L+1}) + \Delta\boldsymbol{G}_{t,L+1})\boldsymbol{p}_{t,L}) = f(\mathrm{Init}(\boldsymbol{G}_{t,L+1})\boldsymbol{p}_{t,L}). \tag{21}$$

Here, the second equality holds since $\boldsymbol{p}_{t,L} \in \mathcal{M}_{t,L+1}$ and $\Delta\boldsymbol{G}_{t,L+1}\perp\mathcal{M}_{t,L+1}$. $\qquad\square$

# B More Details of Experimental Settings

## B.1 More Details of Datasets

Table 5 and Table 6 show the details of Long Sequence Benchmark and SuperNI Benchmark, respectively. Long Sequence Benchmark consists of 15 classification tasks while SuperNI Benchmark consists of various NLP tasks, including dialogue generation, information extraction, question answering, summarization, and sentiment analysis.

Table 5: Details of different tasks in Long Benchmark.

| Dataset name | Category | Domain | Task Type | Metric |
|---|---|---|---|---|
| Yelp | CL Benchmark | sentiment analysis | Yelp reviews | Accuracy |
| Amazon | CL Benchmark | sentiment analysis | Amazon reviews | Accuracy |
| DBpedia | CL Benchmark | topic classification | Wikipedia | Accuracy |
| Yahoo | CL Benchmark | topic classification | Yahoo Q&A | Accuracy |
| AG News | CL Benchmark | topic classification | news | Accuracy |
| MNLI | GLUE | natural language inference | various | Accuracy |
| QQP | GLUE | paraphrase detection | Quora | Accuracy |
| RTE | GLUE | natural language inference | news, Wikipedia | Accuracy |
| SST-2 | GLUE | sentiment analysis | movie reviews | Accuracy |
| WiC | SuperGLUE | word sense disambiguation | lexical databases | Accuracy |
| CB | SuperGLUE | natural language inference | various | Accuracy |
| COPA | SuperGLUE | question and answering | blogs, encyclopedia | Accuracy |
| BoolQA | SuperGLUE | boolean question and answering | Wikipedia | Accuracy |
| MultiRC | SuperGLUE | question and answering | various | Accuracy |
| IMDB | SuperGLUE | sentiment analysis | movie reviews | Accuracy |

The task sequences are constructed using Long Sequence Benchmark and SuperNI Benchmark. The details of different task sequences are presented in Table 7.

## B.2 More Implementation Details

Following existing CL works [40, 61, 68], all methods are implemented using instruction tuning [40]. Experiments are conducted on 5 NVIDIA RTX A6000 GPUs with AdamW [35] as the optimizer. The type of CPU is Intel(R) Xeon(R) Gold 6240R CPU @ 2.40GHz. For T5-Large and T5-XL, their relatively smaller model sizes allow experiments to be performed on a single A6000 GPU with

Table 6: Details of different tasks in SuperNI Benchmark.

| Dataset name | Task Type | Metric |
|---|---|---|
| Task639_multi_woz_user_utterance_generation | summarization | Rouge-L |
| Task1590_diplomacy_text_generation | summarization | Rouge-L |
| Task1729_personachat_generate_next | summarization | Rouge-L |
| Task181_outcome_extraction | information extraction | Rouge-L |
| Task748_glucose_reverse_cause_event_detection | information extraction | Rouge-L |
| Task1510_evalution_relation_extraction | information extraction | Rouge-L |
| Task002_quoref_answer_generation | dialogue generation | Rouge-L |
| Task073_commonsenseqa_answer_generation | dialogue generation | Rouge-L |
| Task591_sciq_answer_generation | dialogue generation | Rouge-L |
| Task511_reddit_tifu_long_text_summarization | question answering | Rouge-L |
| Task1290_xsum_summarization | question answering | Rouge-L |
| Task1572_samsum_summary | question answering | Rouge-L |
| Task363_sst2_polarity_classification | sentiment analysis | Accuracy |
| Task875_emotion_classification | sentiment analysis | Accuracy |
| Task1687_sentiment140_classification | sentiment analysis | Accuracy |

Table 7: The order of different task sequences for experiments.

| Benchmark | Order | Task Sequence |
|---|---|---|
| SuperNI Benchmark | 1 | task1572 → task363 → task1290 → task181 → task002 → task1510 → task639 → task1729 → task073 → task1590 → task748 → task511 → task591 → task1687 → task875 |
| | 2 | task748 → task073 → task1590 → task639 → task1572 → task1687 → task591 → task363 → task1510 → task1729 → task181 → task511 → task002 → task1290 → task875 |
| CL Benchmark | 3 | MNLI → CB → WiC → COPA → QQP → BoolQA → RTE → IMDB → Yelp → Amazon → SST-2 → DBpedia → AG News → MultiRC → Yahoo |
| | 4 | Yelp → Amazon → MNLI → CB → COPA → QQP → RTE → IMDB → SST-2 → DBpedia → AG News → Yahoo → MultiRC → BoolQA → WiC |

gradient accumulation. For Llama-2-7B and Llama-2-13B, data parallelism with DeepSpeed ZeRO-2 [44] is prioritized across multiple A6000 GPUs. FlashAttention-2 [9] is employed to reduce memory usage during training, ensuring sufficient GPU memory to enable DeepSpeed ZeRO-2 whenever possible. However, if the sequence lengths of certain tasks are too long to enable DeepSpeed ZeRO-2 even with FlashAttention-2, DeepSpeed ZeRO-3 is utilized to handle these tasks.

To ensure fair comparisons, for all the methods based on LoRA, we follow existing CL methods [21, 61, 75] by integrating the LoRA architecture into the query and value components of the multi-head attention mechanism in each Transformer block. Following existing works [61, 75], for all the methods based on LoRA, the rank of a single LoRA branch is set to 4 for Order 1 and Order 2, and 8 for Order 3 and Order 4. We also vary the rank in LoRA branches and show the results in Appendix C.4.

For our methods, the global batch size is set to 32 across all model backbones. The learning rate is set to 3e-4 for T5 backbones and 5e-5 for Llama backbones. Each task is trained for 100 epochs with T5 backbones and 50 epochs with Llama backbones. For baselines, we follow their official implementations to set the hyperparameters, making the comparison as fair as possible. If this does not achieve the expected performance, we perform a hyperparameter search for the learning rate and batch size.

### B.3 More Details about the Architecture of the Gating Module

The architecture of the gating module $g_i(\cdot)$ can be represented as

$$
\begin{aligned}
g_i(\boldsymbol{x}) &= f(\boldsymbol{G}_{i,L+1}\boldsymbol{p}_{i,L}), \\
\boldsymbol{p}_{i,l} &= \sigma(\boldsymbol{G}_{i,l}\boldsymbol{p}_{i,l-1}), \; l \in \{1,2,...,L\}, \\
\boldsymbol{p}_{i,0} &= \boldsymbol{p}_0 = \text{Pool}(\text{Token}(\boldsymbol{x})).
\end{aligned}
\tag{22}
$$

Non-linear activation function $\sigma(\cdot)$ is set to SiLU [13]. For all experiments, unless otherwise stated, $L$ is set to 2. In other words, the gating module $g_i(\cdot)$ has three layers. For T5-Large and T5-XL, the parameters in the $i$-th gating module $g_i(\cdot)$ are $\boldsymbol{G}_{i,1} \in \mathbb{R}^{100 \times d}$, $\boldsymbol{G}_{i,2} \in \mathbb{R}^{d \times 100}$ and $\boldsymbol{G}_{i,3} \in \mathbb{R}^{1 \times d}$. For Llama-2-7B and Llama-2-13B, the parameters in the $i$-th gating module $g_i(\cdot)$ are $\boldsymbol{G}_{i,1} \in \mathbb{R}^{50 \times d}$, $\boldsymbol{G}_{i,2} \in \mathbb{R}^{d \times 50}$ and $\boldsymbol{G}_{i,3} \in \mathbb{R}^{1 \times d}$. Here, $d$ denotes the dimension of the embeddings. For different models, $d$ is 1024 for T5-Large and T5-XL, 4096 for Llama-2-7B, and 5120 for Llama-2-13B.

Additionally, we investigate the influence of the architecture of the gating module on the performance of our method. Results are provided in Appendix C.3.

### B.4 Computation of Trainable Parameters

To ensure fair comparisons, we set the same rank for each LoRA branch across all CL methods based on the expandable LoRA architectures shown in Figure 1 (a). Additionally, for all the methods based on LoRA, the LoRA modules are incorporated into the query and value components of the multi-head attention mechanism within each Transformer block.

#### B.4.1 Computation of Trainable Parameters in T5-Large

In T5-Large, the projection weights for the query and value components have shapes $\boldsymbol{W}_q, \boldsymbol{W}_v \in \mathbb{R}^{1024 \times 1024}$. The model consists of 24 self-attention modules in the encoder, 24 self-attention modules in the decoder, and 24 cross-attention modules in the decoder, resulting in a total of $(24 + 24 + 24) * 2 = 144$ pre-trained weights that incorporate the LoRA architecture.

During the learning of the $t$-th new task, O-LoRA updates the parameters $\boldsymbol{A}_t \in \mathbb{R}^{1024 \times r}$ and $\boldsymbol{B}_t \in \mathbb{R}^{r \times 1024}$, resulting in $1024 * r * 144 + r * 1024 * 144 = 294912r$ trainable parameters. When $r = 4$, the number of trainable parameters in O-LoRA is $294912 * 4 = 1179648 = 1.18\text{M}$. InfLoRA only updates the parameters $\boldsymbol{A}_t \in \mathbb{R}^{1024 \times r}$, resulting in $1024 * r * 144 = 147456r$ trainable parameters. When $r = 4$, the number of trainable parameters in InfLoRA is $147456r = 589824 = 0.59\text{M}$.

GainLoRA introduces an additional new gating module $g_t(\cdot)$ with parameters $\boldsymbol{G}_{t,1} \in \mathbb{R}^{100 \times 1024}$, $\boldsymbol{G}_{t,2} \in \mathbb{R}^{1024 \times 100}$ and $\boldsymbol{G}_{t,3} \in \mathbb{R}^{1 \times 1024}$. Therefore, the number of trainable parameters in Gain-LoRA (O-LoRA) is $1179648 + 1024 * 100 + 1024 * 100 + 1024 = 1385472 = 1.39\text{M}$. The number of trainable parameters in GainLoRA (InfLoRA) is $589824 + 1024 * 100 + 1024 * 100 + 1024 = 795648 = 0.80\text{M}$.

#### B.4.2 Computation of Trainable Parameters in T5-XL

In T5-XL, the projection weights for the query and value components have shapes $\boldsymbol{W}_q, \boldsymbol{W}_v \in \mathbb{R}^{4096 \times 1024}$. The model architecture is similar to T5-Large, with 144 pre-trained weights incorporating LoRA.

During the learning of the $t$-th new task, O-LoRA updates the parameters $\boldsymbol{A}_t \in \mathbb{R}^{4096 \times r}$ and $\boldsymbol{B}_t \in \mathbb{R}^{r \times 1024}$, resulting in is $4096 * r * 144 + r * 1024 * 144 = 737280r$ trainable parameters. When $r = 4$, O-LoRA has $737280 * 4 = 2949120 = 2.95\text{M}$ trainable parameters. InfLoRA only updates $\boldsymbol{A}_t \in \mathbb{R}^{4096 \times r}$, resulting in $4096 * r * 144 = 589824r$ trainable parameters. When $r = 4$, InfLoRA has $589824 * 4 = 2359296 = 2.36\text{M}$ trainable parameters.

GainLoRA introduces the same new gating module $g_t(\cdot)$ as in T5-Large, with parameters $\boldsymbol{G}_{t,1} \in \mathbb{R}^{100 \times 1024}$, $\boldsymbol{G}_{t,2} \in \mathbb{R}^{1024 \times 100}$ and $\boldsymbol{G}_{t,3} \in \mathbb{R}^{1 \times 1024}$. Thus, the total number of trainable parameters for GainLoRA (O-LoRA) is $2949120 + 1024 * 100 + 1024 * 100 + 1024 = 3154944 = 3.15\text{M}$. The total number of trainable parameters in GainLoRA (InfLoRA) is $2359296 + 1024 * 100 + 1024 * 100 + 1024 = 2565120 = 2.57\text{M}$.

### B.4.3 Computation of Trainable Parameters in Llama-2-7B

In Llama-2-7B, the projection weights for the query and value components have shapes $W_q, W_v \in \mathbb{R}^{4096 \times 4096}$. The model contains 32 self-attention modules, resulting in $32 * 2 = 64$ pre-trained weights that incorporate the LoRA architecture.

During the learning of the $t$-th new task, O-LoRA updates the parameters $A_t \in \mathbb{R}^{4096 \times r}$ and $B_t \in \mathbb{R}^{r \times 4096}$, resulting in $4096 * r * 64 + r * 4096 * 64 = 524288r$ trainable parameters. When $r = 4$, the number of trainable parameters in O-LoRA is $524288 * 4 = 2097152 = 2.10M$. InfLoRA only updates the parameters $A_t \in \mathbb{R}^{4096 \times r}$, resulting in $4096 * r * 64 = 262144r$ trainable parameters. When $r = 4$, the number of trainable parameters in InfLoRA is $262144 * 4 = 1048576 = 1.05M$.

GainLoRA introduces a new gating module $g_t(\cdot)$ with parameters $G_{t,1} \in \mathbb{R}^{50 \times 4096}, G_{t,2} \in \mathbb{R}^{4096 \times 50}$ and $G_{t,3} \in \mathbb{R}^{1 \times 4096}$. Therefore, the number of trainable parameters in GainLoRA (O-LoRA) is $2097152 + 4096 * 50 + 4096 * 50 + 4096 = 2510848 = 2.51M$. The number of trainable parameters in GainLoRA (InfLoRA) is $1048576 + 4096 * 50 + 4096 * 50 + 4096 = 1462272 = 1.46M$.

### B.4.4 Computation of Trainable Parameters in Llama-2-13B

In Llama-2-13B, the projection weights for the query and value components have shapes $W_q, W_v \in \mathbb{R}^{5120 \times 5120}$. The model contains 40 self-attention modules, resulting in $40 * 2 = 80$ pre-trained weights that incorporate the LoRA architecture.

During the learning of the $t$-th new task, O-LoRA updates the parameters $A_t \in \mathbb{R}^{5120 \times r}$ and $B_t \in \mathbb{R}^{r \times 5120}$, resulting in $5120 * r * 80 + r * 5120 * 80 = 819200r$ trainable parameters. When $r = 4$, the number of trainable parameters in O-LoRA is $819200 * 4 = 3276800 = 3.28M$. InfLoRA only updates the parameters $A_t \in \mathbb{R}^{5120 \times r}$, resulting in $5120 * r * 80 = 409600r$ trainable parameters. When $r = 4$, the number of trainable parameters in InfLoRA is $409600 * 4 = 1638400 = 1.64M$.

GainLoRA introduces a new gating module $g_t(\cdot)$ with parameters $G_{t,1} \in \mathbb{R}^{50 \times 5120}, G_{t,2} \in \mathbb{R}^{5120 \times 50}$ and $G_{t,3} \in \mathbb{R}^{1 \times 5120}$. Therefore, the number of trainable parameters in GainLoRA (O-LoRA) is $3276800 + 5120 * 50 + 5120 * 50 + 5120 = 3793920 = 3.79M$. The number of trainable parameters in GainLoRA (InfLoRA) is $1638400 + 5120 * 50 + 5120 * 50 + 5120 = 2155520 = 2.16M$.

### B.4.5 Computation of Trainable Parameters in Llama-3-8B

In Llama-3-8B, the projection weights for the query and value components have shapes $W_q \in \mathbb{R}^{4096 \times 4096}, W_v \in \mathbb{R}^{4096 \times 1024}$. The model contains 40 self-attention modules, resulting in $32 * 2 = 64$ pre-trained weights that incorporate the LoRA architecture.

During the learning of the $t$-th new task, O-LoRA updates the parameters $A_t \in \mathbb{R}^{4096 \times r}$ and $B_t \in \mathbb{R}^{r \times 4096}$ for query and $A_t \in \mathbb{R}^{1024 \times r}$ and $B_t \in \mathbb{R}^{r \times 4096}$ for value, resulting in $1024 * r * 32 + 4096 * r * 32 + r * 4096 * 64 = 425984r$ trainable parameters. When $r = 4$, the number of trainable parameters in O-LoRA is $425984 * 4 = 1703936 = 1.70M$. InfLoRA only updates the parameters $A_t \in \mathbb{R}^{4096 \times r}$ for query and $A_t \in \mathbb{R}^{1024 \times r}$ for value, resulting in $4096 * r * 32 + 1024 * r * 32 = 163840r$ trainable parameters. When $r = 4$, the number of trainable parameters in InfLoRA is $163840 * 4 = 655360 = 0.66M$.

GainLoRA introduces a new gating module $g_t(\cdot)$ with parameters $G_{t,1} \in \mathbb{R}^{50 \times 4096}, G_{t,2} \in \mathbb{R}^{4096 \times 50}$ and $G_{t,3} \in \mathbb{R}^{1 \times 4096}$. Therefore, the number of trainable parameters in GainLoRA (O-LoRA) is $1703936 + 4096 * 50 + 4096 * 50 + 4096 = 2117632 = 2.12M$. The number of trainable parameters in GainLoRA (InfLoRA) is $655360 + 4096 * 50 + 4096 * 50 + 4096 = 1069056 = 1.07M$.

## C  More Experimental Results

### C.1  Discussing Computational Costs Introduced by GainLoRA

Existing methods, such as O-LoRA and InfLoRA, adopt the expandable LoRA architecture shown in Figure 1 (a) and fix the integration coefficients $\{a_i\}_{i=1}^{T}$ to 1, allowing the model to merge the expanded LoRA branches into the pre-trained matrix at inference time, thereby avoiding additional computational costs. However, when using our gating module to integrate different LoRA branches, the LoRA branches cannot be merged into the pre-trained matrix at inference time, which introduces

Table 8: FLOPs and MACs for different models.

| | Method | Input Shape (batch,length) | FLOPs (G) | MACs (G) |
|---|---|---|---|---|
| T5-Large | Original | (1,128) | 194.25 | 97.1 |
| | GainLoRA (O-LoRA) | (1,128) | 198.79 | 99.37 |
| | GainLoRA (InfLoRA) | (1,128) | 198.79 | 99.37 |
| T5-XL | Original | (1,128) | 751.7 | 375.78 |
| | GainLoRA (O-LoRA) | (1,128) | 763.03 | 381.45 |
| | GainLoRA (InfLoRA) | (1,128) | 763.03 | 381.45 |
| Llama-2-7B | Original | (1,128) | 1701.07 | 850.5 |
| | GainLoRA (O-LoRA) | (1,128) | 1709.14 | 854.53 |
| | GainLoRA (InfLoRA) | (1,128) | 1709.14 | 854.53 |
| Llama-2-13B | Original | (1,128) | 3291.66 | 1645.79 |
| | GainLoRA (O-LoRA) | (1,128) | 3304.26 | 1652.09 |
| | GainLoRA (InfLoRA) | (1,128) | 3304.26 | 1652.09 |
| Llama-3-8B | Original | (1,128) | 1929.86 | 964.89 |
| | GainLoRA (O-LoRA) | (1,128) | 1930.49 | 965.21 |
| | GainLoRA (InfLoRA) | (1,128) | 1930.49 | 965.21 |

additional computational costs. Nevertheless, we demonstrate that these computational costs are minimal compared to the computational cost of the original large language models (LLMs).

Table 8 compares the floating-point operations (FLOPs) and multiply-add operations (MACs) during inference for different models with and without GainLoRA. The computation of FLOPs and MACs follows the existing project calflops [69]. Here, "Original" denotes the original LLMs without any LoRA adaptation. Methods such as O-LoRA and InfLoRA avoid additional computational costs by merging their LoRA branches into the original weights during inference, resulting in FLOPs and MACs identical to the original LLMs. Despite introducing additional FLOPs and MACs compared to the original LLMs, GainLoRA maintains minimal computational overhead relative to the original LLMs.

## C.2 Additional Computation Introduced by Subspace Construction

The memory and computational overhead of subspace construction in GainLoRA is minimal due to the small size of the gating module (only 3 layers, see B.3). We provide detailed analyses below.

**Memory** The number of orthogonal bases stored for each subspace does not exceed its dimension. For T5-Large, the dimensions of the three subspaces are 1024, 100, and 1024, respectively. This results in a worst-case memory of less than 0.3% of the total model parameters ($(2*1024^2 + 100^2)$/(T5-Large's params)<0.3%). Similar estimates yield 0.07%, 0.5%, and 0.4% for T5-XL, Llama-2-7B, Llama-2-13B and Llama-3-8B, respectively. Since this calculation represents a rough upper bound, the actual memory is even lower.

**Computational Overhead** The computational overhead for subspace construction requires a single forward pass over the task dataset and SVD on the feature matrices of the gating module.

Assuming a single forward pass over the task dataset requires $A$ FLOPs. For T5-Large, training a task for 100 epochs needs 100 forward and backward passes. Since a single backward pass has roughly $2A$ FLOPs, the total FLOPs are $500A$. Thus, a single forward pass for subspace construction accounts for only 1/500=0.2% of total computation. Similar estimates yield 0.2%, 0.4%, and 0.4% for T5-XL, Llama-2-7B, and Llama-2-13B, respectively.

GPM requires performing SVD on matrix $H_l H_l^T \in \mathbb{R}^{d_l \times d_l}$, where $H_l$ is the feature matrix in the $l$-th layer of gating module. Based on existing conclusions [55], the FLOPs for SVD on $HH^T$ is less than $10d_l^3$. For T5-Large ($d_1 = d_3 = 1024$ and $d_2 = 100$), this results in $10 * (2*1024^3 + 100^3) < 30 GFLOPs$, which is negligible compared to a single forward pass with sequence length 128 (see Table 8). Similar calculations give the same conclusion for T5-XL, Llama-2-7B, Llama-2-13B and Llama-3-8B.

## C.3 Varying the Architecture of Gating Module

Table 9: Varying the function $f(\cdot)$ in GainLoRA on different task sequences with T5-large model.

| Method | Order 1 | | Order 2 | |
|---|---|---|---|---|
| | AP↑ | FT↓ | AP↑ | FT↓ |
| GainLoRA (InfLoRA) $(f(b) = \|2\text{sigmoid}(b) - 1\|)$ | 46.21 | 2.40 | **46.44** | 2.61 |
| GainLoRA (InfLoRA) $(f(b) = \min\{\|b\|, 1\})$ | 45.05 | 2.07 | 45.00 | **1.74** |
| GainLoRA (InfLoRA) $\left(f(b) = \|\sin(\frac{\pi b}{2})\|\right)$ | **47.48** | **1.21** | 45.03 | 2.37 |
| InfLoRA | 39.78 | 7.64 | 39.57 | 8.93 |
| GainLoRA (O-LoRA) $(f(b) = \|2\text{sigmoid}(b) - 1\|)$ | 47.84 | **2.26** | 46.84 | **2.91** |
| GainLoRA (O-LoRA) $(f(b) = \min\{\|b\|, 1\})$ | **49.62** | 2.83 | **48.62** | 3.74 |
| GainLoRA (O-LoRA) $\left(f(b) = \|\sin(\frac{\pi b}{2})\|\right)$ | 48.49 | 3.84 | 47.20 | 4.69 |
| O-LoRA | 26.37 | 19.15 | 32.83 | 11.99 |

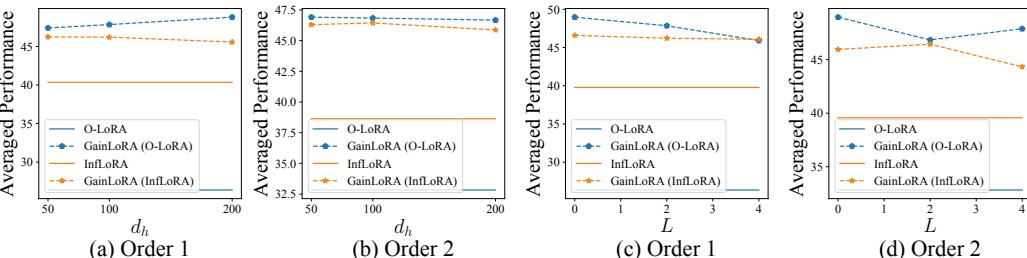

Figure 5: (a) and (b) show the variation of our methods' performance with the shapes of the weights in the gating module. (c) and (d) show the variation of our methods' performance with the Layers of the gating module.

### C.3.1 Varying Function $f(\cdot)$ in Gating Module

To implement our method, we define function $f(\cdot)$ as (8). Here, we vary the formula of function $f(\cdot)$ as the following two functions:

$$\min\{|b|, 1\}, \; |\sin(\frac{\pi b}{2})|. \tag{23}$$

Clearly, these two functions map real values among $[0, 1]$ and satisfy $f(0) = 0$. Table 9 shows the results. As we can see, when changing the formula of $f(\cdot)$, GainLoRA also improves the performance of O-LoRA and InfLoRA.

### C.3.2 Varying the Shapes of Weights in Gating Module

In this section, we vary the shapes of the weights in the gating modules with T5-Large. Specifically, we set the weights $\boldsymbol{G}_{i,1} \in \mathbb{R}^{d_h \times 1024}$ and $\boldsymbol{G}_{i,2} \in \mathbb{R}^{1024 \times d_h}$ in each gating module $g_i(\cdot)$ and vary $d_h$ over $\{50, 100, 200\}$. Figure 5 (a) and Figure 5 (b) show the results. As we can see, when increasing $d_h$, the performance of GainLoRA remains relatively stable, indicating that our method is robust to the shape of the weights in the gating module. Note that the number of trainable parameters increases as $d_h$ increases.

### C.3.3 Varying the Layers of Gating Module

In this section, we vary the layers of the gating modules with T5-Large. Specifically, we vary across $\{0, 2, 4\}$. when $L = 0$, there is only one layer with $\boldsymbol{G}_{i,1} \in \mathcal{R}^{1 \times 1024}$ in each gating module $g_i(\cdot)$. When $L = 2$, there are three layers with $\boldsymbol{G}_{i,1} \in \mathcal{R}^{100 \times 1024}$, $\boldsymbol{G}_{i,2} \in \mathcal{R}^{1024 \times 100}$ and $\boldsymbol{G}_{i,3} \in \mathcal{R}^{1 \times 1024}$. When $L = 4$, there are 5 layers with $\boldsymbol{G}_{i,1} \in \mathcal{R}^{100 \times 1024}$, $\boldsymbol{G}_{i,2} \in \mathcal{R}^{1024 \times 100}$, $\boldsymbol{G}_{i,3} \in \mathcal{R}^{100 \times 1024}$, $\boldsymbol{G}_{i,4} \in \mathcal{R}^{1024 \times 100}$, and $\boldsymbol{G}_{i,5} \in \mathcal{R}^{1 \times 1024}$ in each gating module. Figure 5 (c) and Figure 5 (d) show the results. As we can see, when increasing the layers of gating modules, the performance of GainLoRA remains relatively stable, indicating that our method is robust to the layers of the gating module. Note that the number of trainable parameters increases as the number of layers in gating modules increases.

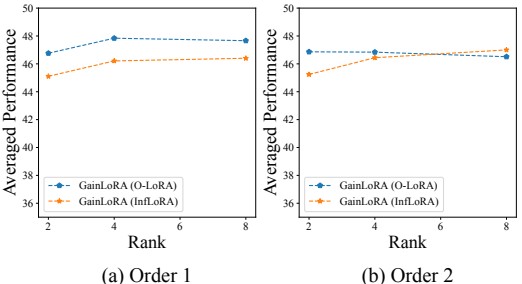

| | (a) Order 1 | (b) Order 2 |

Figure 6: The variation of our methods' performance with the Layers of the gating module.

Table 10: The overall results on different task sequences with T5-large model.

| Method | Order 1 | | Order 2 | |
| | AP↑ | FT↓ | AP↑ | FT↓ |
|---|---|---|---|---|
| IncLoRA | 12.33 | 41.93 | 16.65 | 36.56 |
| GainLoRA (IncLoRA) | 47.82 | 3.73 | 45.42 | **5.83** |
| C-LoRA | 22.69 | 24.25 | 32.81 | 11.60 |
| GainLoRA (C-LoRA) | **49.24** | **2.94** | **46.23** | 6.05 |

## C.4 Varying Ranks in LoRA Branches

In this section, we vary the rank of LoRA branches across $\{2, 4, 8\}$ with T5-Large. Figure 6 shows the results. As shown, when the rank of LoRA branches increases, the performance of GainLoRA remains relatively stable. Note that the number of trainable parameters increases as the rank of LoRA branches increases.

## C.5 Adopting Other Update Strategies for the New LoRA Branch

Our GainLoRA does not impose specific update strategies for the new LoRA branches. In this work, we adopt the same update strategies as the existing two methods, O-LoRA [61] and InfLoRA [32]. Related methods, such as IncLoRA [21] and C-LoRA [50], also adopt the expandable LoRA architecture illustrated in Figure 1 (a) and fix all integration coefficients $\{a_i\}_{i=1}^{T}$ to 1. Our method GainLoRA can also adopt their update strategies for the new LoRA branch. Table 10 presents the results, demonstrating that GainLoRA further improves the performance of these two methods.

## C.6 Performance on the TRACE Benchmark

To further demonstrate our method's effectiveness, we follow existing CL methods for LLMs [18, 62] and conduct experiments on the TRACE dataset [62] with Llama-2-7B-Chat. The dataset comprises a diverse set of challenging instruction-tuned tasks, spanning multilingual comprehension, domain-specific knowledge, arithmetic reasoning, and coding. An overview of the tasks in TRACE is provided in Table 11.

Table 11: The order of TRACE benchmark for experiments.

| Benchmark | Task Sequence |
|---|---|
| TRACE Benchmark | C-STANCE → FOMC → MeetingBank → Py150 → ScienceQA → NumGLUE-cm → NUMGLUE-ds → 20Minuten |

Table 12 reports the average performance on the TRACE benchmark after sequentially learning all tasks. The results demonstrate that our method effectively mitigates catastrophic forgetting and outperforms existing baselines. This capability is crucial for real-world applications.

**Retention of General Capabilities** We also follow existing work [62] to explicitly evaluate the preservation of general abilities, such as instruction-following, after continual learning on the TRACE

Table 12: TRACE benchmark performance using LLama-2-7B-Chat.

|  | AP↑ | FT↓ |
|---|---|---|
| O-LoRA [61] | 41.04 | 8.05 |
| GainLoRA (O-LoRA) | 48.10 | 0.99 |
| InfLoRA [32] | 47.67 | 2.25 |
| GainLoRA (InfLoRA) | 49.15 | 0.89 |

Table 13: Comparison of general ability scores across six diverse evaluation tasks between the base LLaMA-2-7B chat model and different methods.

|  | PIQA | MMLU | GSM8K | BBH | BoolQA | TydiQA |
|---|---|---|---|---|---|---|
| O-LoRA [61] | 72.85 | 32.87 | 13.42 | 35.10 | 56.88 | 19.48 |
| GainLoRA (O-LoRA) | 73.61 | 33.33 | 18.57 | 36.47 | 59.69 | 25.00 |
| InfLoRA [32] | 74.86 | 40.86 | 15.69 | 35.87 | 65.29 | 27.25 |
| GainLoRA (InfLoRA) | 75.24 | 44.25 | 21.30 | 37.44 | 68.81 | 27.84 |
| Llama-2-7B-Chat | 75.35 | 46.13 | 26.54 | 40.09 | 70.46 | 23.45 |

benchmark using different methods. Table 13 indicates that continual learning with different methods often leads to a degradation of general abilities. However, GainLoRA demonstrates a stronger ability to mitigate forgetting compared to other LoRA-based methods, including O-LoRA and InfLoRA.

## C.7 Compared with More CL Methods in CV

Following many existing continual learning methods in NLP [75, 61, 18], this paper focuses on models based on next-token prediction, such as T5 [43] and LLaMA [54], which lack the [CLS] token used in ViT. Although many continual learning methods based on pre-trained models in computer vision [66, 67, 63, 57, 58, 51] cannot be directly applied to our setting, we adapt several of them to the T5 architecture to ensure a comprehensive comparison. Specifically, We implement these methods by injecting prompts into both the keys and values in T5, and introduce zero-padding in the position bias tensor to ensure shape compatibility. Note that we do not add positional information to the prompts, which is consistent with DualPrompt and CODA-Prompt in ViT. The results on order 1 with T5 are reported in Table 14, showing that these methods perform significantly worse than our GainLoRA and exhibit noticeable forgetting.

Table 14: Compare with different methods on order 1 with T5 architecture.

|  | AP↑ | FT↓ |
|---|---|---|
| L2P [67] | 15.23 | 11.34 |
| DualPrompt [66] | 17.40 | 10.63 |
| CODA-Prompt [51] | 19.28 | 14.62 |
| GainLoRA (O-LoRA) | **47.84** | **2.26** |
| GainLoRA (InfLoRA) | 46.21 | 2.40 |

