# OpenReview forum: "Gated Integration of Low-Rank Adaptation for Continual Learning  of Large Language Models"
_NeurIPS.cc/2025/Conference — NeurIPS 2025 poster_

### Official Review · Reviewer_fYm8 · 2025-06-28

**Clarity:** 2
**Significance:** 2
**Originality:** 3
**Rating:** 4
**Confidence:** 3

**Summary:**

This article proposes a method called GainLoRA for continuous learning of language models. This method aims to solve the problem that existing LoRA-based CL methods easily forget old task knowledge when learning new tasks. By extending a new LoRA branch for each new task and introducing a gating module to integrate the new and old LoRA branches, the new gating module is used to minimize the contribution of the new LoRA branch to the old task, thereby effectively alleviating forgetting and improving the overall performance of the model.

**Questions:**

1、The GainLoRA method proposed in the article was only experimented on T5 and Llama2. Can more comprehensive experiments be conducted on the latest models such as Qwen3 to verify the generalization of the method?
2、Have the authors considered verifying the effectiveness of GainLoRA in more complex scenarios, such as scenarios where there is overlap between tasks, rather than just sequentially learning non-overlapping tasks?

**Ethical Concerns:**

["NO or VERY MINOR ethics concerns only"]

**Limitations:**

yes

**Paper Formatting Concerns:**

No major formatting issues were found in this paper.

**Quality:**

2

**Strengths And Weaknesses:**

Strengths：
1、The article proposes an innovative gating module for integrating new and old LoRA branches. This gating mechanism can dynamically adjust the contribution of the new LoRA branch to the old task, and by imposing specific initialization and update constraints, the impact of the new LoRA branch on the old task when learning the new task is minimized.
2、The GainLoRA method has good compatibility and can be combined with a variety of existing LoRA-based CL methods, and can be extended to language model architectures of different sizes.

Weaknesses：
1、The GainLoRA method proposed in the article was only experimented on T5 and Llama2, and lacked experiments on the latest models such as Qwen3.
2、In GainLoRA, the initialization constraint of the new gating module requires it to be orthogonal to the subspace of the previous t-1 tasks. However, this may cause the model to fail to fully utilize the useful information related to the old tasks in the initial stage of the new task, thereby affecting the learning of the new task or making the model unable to transfer the knowledge of the old task to the new task.

---

> ### Author Rebuttal · Authors · 2025-07-31
>
> **Q1: The GainLoRA method proposed in the article was only experimented on T5 and Llama2, and lacked experiments on the latest models such as Qwen3.**
>
> **A1** In the table below, we present the experimental results of our method on Order 1 using Qwen-3-8B and LLaMA-3-8B. As shown, all methods perform better with Qwen-3-8B and LLaMA-3-8B compared to LLaMA-2-7B. Moreover, our method, GainLoRA, achieves the best performance on both Qwen-3-8B and LLaMA-3-8B. We will include these results in the final version of the paper. Thank you for the suggestion.
>
> |Qwen-3-8B|AP|FT
> :-|:-:|:-:
> OLoRA|44.11|8.85
> GainLoRA (OLoRA)|**54.28**|2.05
> InfLoRA|45.69|9.43
> GainLoRA (InfLoRA)|53.59|**1.80**
>
> |Llama-3-8B|AP|FT
> :-|:-:|:-:
> OLoRA|42.49|12.85
> GainLoRA (OLoRA)|**53.39**|3.56
> InfLoRA|43.27|6.02
> GainLoRA (InfLoRA)|52.18|**1.40**
>
>
> **Q2: In GainLoRA, the initialization constraint of the new gating module requires it to be orthogonal to the subspace of the previous t-1 tasks. However, this may cause the model to fail to fully utilize the useful information related to the old tasks in the initial stage of the new task, thereby affecting the learning of the new task or making the model unable to transfer the knowledge of the old task to the new task.**
>
> **A2** We selected three sentiment analysis tasks (Task363, Task1687, Task875) from the 15 tasks in Order1 and Order2. Intuitively, these tasks belong to the same type and are therefore more similar to each other than to tasks from a different type. We evaluated GainLoRA on these tasks (Task363 → Task1687 → Task875). The results in the table show that while GainLoRA remains effective, its improvement is smaller than that in the 15-task setting with dissimilar tasks. Furthermore, GainLoRA underperforms InfLoRA and O-LoRA on the new task (Task875) but outperforms them on the old tasks (Task363 and Task1687). This suggests that orthogonality constraints might hinder forward transfer. This is a common trade-off in continual learning (CL): constraints help mitigate forgetting but may restrict transfer, particularly for similar tasks. Conversely, weak or no constraints increase the risk of forgetting in dissimilar tasks. Future work will explore adaptive strategies: stronger constraints for dissimilar tasks and weaker ones for similar tasks. Anyway, our GainLoRA is effective in terms of overall (average) accuracy. These discussions and results will be included in the final version of the paper. Thanks for this insightful comment.
>
> ||Task363|Task1687|Task875|Avg
> :-|:-:|:-:|:-:|:-:
> OLoRA|85.0|75.3|**79.7**|80.0
> GainLoRA (OLoRA)|90.7|79.3|77.3|**82.4**
> InfLoRA|88.3|75.7|**79.3**|81.1
> GainLoRA (InfLoRA)|**92.0**|78.7|76.3|82.3
>
>
> **Q3: Have the authors considered verifying the effectiveness of GainLoRA in more complex scenarios, such as scenarios where there is overlap between tasks, rather than just sequentially learning non-overlapping tasks?**
>
> **A3** While we consider some alternative scenarios such as the similar tasks discussed in **A2**, we do not include experiments under explicit task-overlapping settings. This is because the form of task overlapping in NLP needs further investigation and clarification. Specifically, we follow existing NLP continual learning setups [1, 2] and assume a single task sequence may involve diverse task types such as summarization and classification. In this case, it remains unclear whether task overlapping should only occur between tasks of the same type, or if it can also happen across different types of tasks.
>
> We agree that evaluating continual learning methods in overlapping task settings is an important and challenging direction for future research. It requires not only task design but also methodological advances that go beyond catastrophic forgetting mitigation. GainLoRA, as currently designed, focuses on mitigating forgetting and may not be directly effective under overlapping scenarios, which is consistent with the no-free-lunch principle.
>
> In the final version, we will explicitly discuss this limitation and highlight overlapping tasks as a promising direction for future work. Thanks for the thoughtful suggestion.
>
> [1] Wang X, Chen T, Ge Q, et al. Orthogonal Subspace Learning for Language Model Continual Learning, EMNLP 2023.
>
> [2] Zhao W, Wang S, Hu Y, et al. SAPT: A Shared Attention Framework for Parameter-Efficient Continual Learning of Large Language Models, ACL 2024.

---

> > ### Comment · Reviewer_fYm8 · 2025-08-04
> > **Response To Authors**
> >
> > Thank you very much for your detailed response. I have carefully reviewed your replies to the questions I raised. The authors present experimental results on the latest models. From the author's answers to Weakness 2 and Question 2, I got some answers to these two points. However, I still believe my current rating is appropriate, and I will not be adjusting it further. Thank you for your patient response

---

### Official Review · Reviewer_uWep · 2025-06-30

**Clarity:** 4
**Significance:** 3
**Originality:** 3
**Rating:** 4
**Confidence:** 5

**Summary:**

This paper introduces GainLoRA, a parameter‐efficient continual‐learning method for language models that builds on the popular LoRA framework. For each new task, GainLoRA adds a new low‐rank adaptation branch and integrates old and new branches via lightweight, task‐specific gating modules.  By imposing orthogonality constraints at initialization and during updates, GainLoRA drives the new branch’s contribution toward zero on previously learned tasks, thereby mitigating catastrophic forgetting while allowing full expressivity on the current task.  Extensive experiments on SuperNI and Long Sequence benchmarks, across multiple task orders and model scales are conducted.

**Questions:**

See weakness

**Ethical Concerns:**

["NO or VERY MINOR ethics concerns only"]

**Final Justification:**

Thank the authors for providing justifications and additional experiments. Most of my concerns are addressed. I will raise my final rating.

**Limitations:**

yes

**Quality:**

4

**Strengths And Weaknesses:**

Strengths
1. The presentation is straightforward and easy to follow.
2. The experiments and ablations are comprehensive, addressing key concerns.
3. The proposed GainLoRA can be plugged into other existing LoRA-based methods.

Weaknesses
1. The gating module itself is not inherently novel, as similar mechanisms using query-key matching ([1,2,3]) or moe ([4,5]) have been explored in PEFT and CL methods.
2. The proposed gating module is a collection of task-wise 3-layer MLPs with SiLU activation, operates in parallelly to the LLM backbone, which acts as a task-id predictor. Fig. 3(c,d) reports gating activations only for the most recent (last) task. To fully support the paper’s claims, the authors should also report gating outputs (or task‐prediction accuracy) for **each** of the previous tasks.
3. Weird performance in Table 18, two stronger variants of L2P ( DualPrompt and Coda-prompt) both underperform the weaker baseline L2P. The low AP and low FT mean the model fails to learn new knowledge as new tasks are introduced, which suggests that there may exist some errors/bugs in the reimplementation of these methods in the LM domain. As the authors noted, there is no [CLS] token, whereas a similar operation as in Eq.3 should be used for these methods without using the proposed constraints.

[1] Learning to Prompt for Continual Learning https://arxiv.org/abs/2112.08654

[2] DualPrompt: Complementary Prompting for Rehearsal-free Continual Learning https://arxiv.org/abs/2204.04799

[3] S-Prompts Learning with Pre-trained Transformers: An Occam's Razor for Domain Incremental Learning https://arxiv.org/abs/2207.12819

[4] CODA-Prompt: COntinual Decomposed Attention-based Prompting for Rehearsal-Free Continual Learning https://arxiv.org/abs/2211.13218

[5] Self-Expansion of Pre-trained Models with Mixture of Adapters for Continual Learning https://arxiv.org/abs/2403.18886

---

> ### Author Rebuttal · Authors · 2025-07-31
>
> **Q1: The gating module itself is not inherently novel, as similar mechanisms using query-key matching ([1,2,3]) or moe ([4,5]) have been explored in PEFT and CL methods.**
>
> **A1** We acknowledge that the gating module itself is not new, and we do not claim novelty in the structure of the gating module itself. However, what differentiates our approach is the application scenarios, motivation and technical design, particularly within the setting of continual learning (CL) for NLP using LoRA.
>
> First, our work focuses on NLP scenarios, but the methods mentioned by the reviewer primarily focus on CV tasks. Due to differences in model architectures and datasets between NLP and CV, prior works [1, 2] show that CL methods being effective in CV may not perform well in NLP. This supports the “no free lunch”principle.
>
> Second, our motivation is different from existing methods. We show that existing NLP continual learning methods using multiple LoRA branches typically sum them in a naïve and static manner, potentially leading to forgetting. Our method addresses this issue by introducing gating mechanism to dynamically weigh different LoRA branches. This is a different motivation compared to query-key matching methods or MoE methods.
>
> Third, our technical design introduces orthogonal projection for both the initialization and update of the gating module. This ensures that newly learned gating parameters do not interfere with those from previous tasks. To the best of our knowledge, no methods have attempted to impose orthogonal projections as constraints on both the initialization and updating of the gating module.
>
> We have already cited query-key matching methods and will include additional citations to MoE-based methods mentioned by the reviewer in the final version. We will also clarify these differences more explicitly in the paper, particularly in terms of application scenarios, motivation, and technical design. Thanks for the valuable suggestions.
>
>
> **Q2: The proposed gating module is a collection of task-wise 3-layer MLPs with SiLU activation, operates in parallelly to the LLM backbone, which acts as a task-id predictor. Fig. 3(c,d) reports gating activations only for the most recent (last) task. To fully support the paper’s claims, the authors should also report gating outputs (or task‐prediction accuracy) for each of the previous tasks.**
>
> **A2** We would like to clarify that the goal of the proposed gating module is not to predict the task identity. As shown in Figure 1 of the main text, each LoRA branch has its own gating module, and these gating modules are responsible for determining the weights of their corresponding LoRA branches based on the input samples. The objective of the gating module is to ensure that the output for the current task $t$ is close to 1, and for previous tasks $s$ ($1\leq s< t$), the output is close to 0. For the input samples from future tasks $u$ ($u>t$), no specific constraint is imposed on the $t$ gating module, since the gating module for task $t$ will not cause any forgetting for future task $u$. This design further distinguishes the gating module from a task-id predictor.
>
> Fig. 3(c,d) reports the outputs of the $15$-th gating module across both the 15-th tasks and all previous tasks (task ID<15). In the table below, we report (mean, std) for the outputs of the $t$-th gating modules across old tasks (task ID$<t$) and the current task $t$ on order 1, where $1<t<15$. As shown, gating modules consistently output values close to 1 for samples of the current task and values close to 0 for samples of all previous tasks. We will incorporate these results into the final version in a similar graphical form as Fig. 3(c,d). Thanks for the suggestions.
>
> |GainLoRA (OLoRA)|2|3|4|5|6|7|8|9|10|11|12|13|14
> :-|:-:|:-:|:-:|:-:|:-:|:-:|:-:|:-:|:-:|:-:|:-:|:-:|:-:
> old tasks (task ID$<t$) |(0.14,0.08)|(0.26,0.05)|(0.20,0.10)|(0.22,0.11)|(0.13,0.07)|(0.16,0.09)|(0.12,0.09)|(0.10,0.05)|(0.19,0.09)|(0.12,0.07)|(0.15,0.10)|(0.06,0.04)|(0.15,0.07)
> current task $t$ |(0.94,0.00)|(0.94,0.03)|(0.95,0.01)|(0.93,0.01)|(0.96,0.00)|(0.94,0.00)|(0.95,0.00)|(0.94,0.01)|(0.95,0.00)|(0.92,0.02)|(0.95,0.01)|(0.95,0.01)|(0.92,0.01)
>
>
> **Q3: Weird performance in Table 18, two stronger variants of L2P (DualPrompt and Coda-prompt) both underperform the weaker baseline L2P. The low AP and low FT mean the model fails to learn new knowledge as new tasks are introduced, which suggests that there may exist some errors/bugs in the reimplementation of these methods in the LM domain. As the authors noted, there is no [CLS] token, whereas a similar operation as in Eq.3 should be used for these methods without using the proposed constraints.**
>
> **A3** We would like to clarify that the performance in Table 18 is not due to any bugs in our reimplementation. Instead, it is primarily due to architectural differences between ViT and T5, which make it non-trivial to directly apply prompt-based methods originally designed for ViT (vision Transformer) to language models such as T5.
>
> Specifically, in our initial implementation of prompt-based methods, we injected prompts into the input embeddings rather than into the keys and values of the attention layers as official implementation of many prompt-based methods in CV. This design choice was made because T5 uses relative position encoding by adding a position bias to the attention scores, while ViT adds position embeddings directly into the input. Injecting prompts to the keys of some Transformer layers increases the attention score length in these layers, causing a shape mismatching with the position bias in T5.
>
> We re-implemented these methods by injecting prompts into both the keys and values, and introduced zero-padding in the position bias tensor to ensure shape compatibility. Note that we did not add positional information to the prompts, which is consistent with DualPrompt and CODA-Prompt in ViT. Furthermore, as suggested by the reviewer, we generate the query using a 3-layer MLP. The updated results are reported below, showing that both DualPrompt and CODA-Prompt now outperform L2P.
>
> ||AP|FT
> :-|:-:|:-:
> L2P|15.23|11.34|
> DualPrompt|17.40|10.63
> CODA-prompt|19.28|14.62
>
> In addition, we would like to emphasize that the low AP and FT are consistent with prior reproduction efforts for L2P in the LM domain [1] and are not unexpected. Specifically, a recent study [2] has shown that prompt-based methods converge more slowly than LoRA and typically require larger learning rates to be effective. However, in continual learning for LMs, where methods like L2P, DualPrompt, and CODA-Prompt struggle with catastrophic forgetting, using large learning rates can introduce instability. Hence, the learning rate cannot be set to be too large, which leads to relatively low AP and FT. If the learning rate is set to be too large, it will result in a lower AP and a larger FT.
>
> We will add these results and discuss the key techniques and difficulties for adapting methods from CV to NLP in the final version. Thanks a lot for the insightful comment.
>
> [1] Zhao W, Wang S, Hu Y, et al. SAPT: A Shared Attention Framework for Parameter-Efficient Continual Learning of Large Language Models, ACL 2024.
>
> [2] Gao Q, Zhao C, Sun Y, et al. A unified continual learning framework with general parameter-efficient tuning. ICCV, 2023.

---

### Official Review · Reviewer_5sqb · 2025-07-04

**Clarity:** 3
**Significance:** 3
**Originality:** 3
**Rating:** 5
**Confidence:** 3

**Summary:**

In this paper, the authors introduce a novel method to mitigate catastrophic forgetting in continual learning (CL) using language models (LMs). Specifically, the authors introduce gated integration of LoRA (GainLoRA), which will adaptively control the output of a set of LoRA modules corresponding to different tasks such that they do not interfere with each other. GainLoRA achieves this by imposing constraints on the initialization and the update of the gating module.  Thus, this approach does not require samples from prior tasks or task labellings to mitigate the interference of the LoRA adapters that are irrelevant to a given task at the inference time. The empirical results provided in the paper suggest that incorporating the proposed gating mechanism with the existing LoRA-based CL methods can improve their ability to mitigate catastrophic forgetting and improve the average accuracy across tasks.

**Questions:**

* Can the authors provide more discussion/explanation on why the updates (9) and (10) allow the initialization and the update to satisfy the constraints (5) and (6), respectively?

* How can the gating model assign the correct weight to the LoRA output when the input distribution is similar accross the tasks, whie the output is different?

**Ethical Concerns:**

["NO or VERY MINOR ethics concerns only"]

**Final Justification:**

The authors were able to address my concerns in the rebuttal period, and agreed to revise the final version based on these concerns. Thus, I recommend this paper for acceptence.

**Limitations:**

Please refer to the questions above.

**Quality:**

3

**Strengths And Weaknesses:**

**Strengths**

* The paper discusses the problem of catastrophic forgetting in LMs in the CL setting, which is a critical and pertinent problem in the field of machine learning.
* The flow of the paper is easy to follow, and the proposed method is introduced intuitively.
* The ablation studies provided in the paper validate the importance of the different components of the proposed method
* The empirical results provided in the paper suggest that the performance of existing methods designed for CL can be significantly improved by the proposed method.

**Weaknesses**

* Some key information needed to follow the paper seems to be missing in the main text. For example, some more explanation on why the operation in (9) makes the initialization to satisfy the constraint (5), in addition to providing a reference to prior work.

* It is unclear how only input can be used to determine the identity of the task. For example, the same input can be used for a text generation task (e.g., summarization) and a classification task. How can the proposed method mitigate the task interference in a situation like this?

**Minor Weaknesses**

* Line 127: $p_{L+1}$ -> $p_{L}$?
* Although the main text points to the supplementary for some key definitions and metrics, the supplementary is not provided adjacent to the main text, which makes the paper less self-contained.

---

> ### Author Rebuttal · Authors · 2025-07-31
>
> **Q1: Some key information needed to follow the paper seems to be missing in the main text. For example, some more explanation on why the operation in (9) makes the initialization to satisfy the constraint (5), in addition to providing a reference to prior work.**
>
> **A1** Thank you for pointing this out. Here, we provide a more detailed explanation to clarify this point.
>
> The operation in (9) is designed to project the initialization ${\rm Init}(\mathbf{G}\_{t,L+1})$ onto the orthogonal complement of the subspace $\mathcal{M}\_{t,L+1}$, which is spanned by the columns of the orthonormal basis matrix $\mathbf{M}\_{t,L+1}$. Specifically, the projection matrix in (9) is given by
> $${\rm Init}(\mathbf{G}\_{t,L+1})-\mathbf{M}\_{t,L+1}\mathbf{M}\_{t,L+1}^{T}{\rm Init}(\mathbf{G}\_{t,L+1})=(\mathbf{I}-\mathbf{M}\_{t,L+1}\mathbf{M}\_{t,L+1}^{T}){\rm Init}(\mathbf{G}\_{t,L+1}).$$
> For any vector $\mathbf{m}\in\mathcal{M}\_{t,L+1}$, since $\mathcal{M}\_{t,L+1}$ is spanned by the columns of the orthonormal basis matrix $\mathbf{M}\_{t,L+1}$, we have $\mathbf{m}=\mathbf{M}\_{t,L+1}\mathbf{a}$ for some coefficient vector $\mathbf{a}$. Then
> $$\mathbf{m}^{T}({\rm Init}(\mathbf{G}\_{t,L+1})-\mathbf{M}\_{t,L+1}\mathbf{M}\_{t,L+1}^{T}{\rm Init}(\mathbf{G}\_{t,L+1}))=\mathbf{a}^{T}\mathbf{M}\_{t,L+1}^{T}(\mathbf{I}-\mathbf{M}\_{t,L+1}\mathbf{M}\_{t,L+1}^{T}){\rm Init}(\mathbf{G}\_{t,L+1})=\mathbf{a}^{T}(\mathbf{I}-\mathbf{M}\_{t,L+1}^{T}\mathbf{M}\_{t,L+1})\mathbf{M}\_{t,L+1}^{T}{\rm Init}(\mathbf{G}\_{t,L+1}).$$
> Since the columns of $\mathbf{M}\_{t,L+1}$ form an orthonormal basis, we have  $\mathbf{M}\_{t,L+1}^{T}\mathbf{M}\_{t,L+1}=\mathbf{I}$, which means $\mathbf{I}-\mathbf{M}\_{t,L+1}^{T}\mathbf{M}\_{t,L+1}=\mathbf{O}$. Therefore, the above equation is equal to zero matrix $\mathbf{O}$, which means the projected initialization in (9) is orthogonal to the entire space $\mathcal{M}\_{t,L+1}$, and hence the constraint (5) is satisfied. We will add the above explanation in the final version.
>
> **Q2: It is unclear how only input can be used to determine the identity of the task. For example, the same input can be used for a text generation task (e.g., summarization) and a classification task. How can the proposed method mitigate the task interference in a situation like this?**
>
> **A2** We follow the instruction tuning paradigm as adopted in existing works [1, 2]. In particular, even if the same input can be used for a summarization task and a classification task, the corresponding instructions are inherently different. For instance, the instruction for summarization may be: "Summarize the following passage," whereas the instruction for classification task may be: "Classify the following text into one of the predefined categories: (A)..., (B)..., (C)...". In this case, our mechanism can identify different tasks based on the differences in the input instructions.
>
> Without the instruction, neither our method nor other methods can handle both summarization and classification tasks for the same input. Specifically, in the absence of instructions, methods such as OLoRA and TaSL will get the same output for a given input. Hence, in real applications, we adopt different instructions for different tasks.
>
>
> **Q3: Line 127: p_{L+1} -> p_{L}?**
>
> **A3** Yes, this is a typo. We will fix it in the final version.
>
>
> **Q4: Although the main text points to the supplementary for some key definitions and metrics, the supplementary is not provided adjacent to the main text, which makes the paper less self-contained.**
>
> **A4** We sincerely apologize for any inconvenience to the reviewers. To provide more comprehensive support for our work, we have added lots of clarifications and additional experiments in the supplementary materials. We will include the supplementary adjacent to the main text in the final version to ensure the paper is more self-contained and easier to follow.
>
>
> **Q5: Can the authors provide more discussion/explanation on why the updates (9) and (10) allow the initialization and the update to satisfy the constraints (5) and (6), respectively?**
>
> **A5** The discussion in **A1** has shown that the update in (9) allows the initialization to satisfy the constraint in (5). With the same proving process, we can show that the update in (10) allows the update to satisfy the constraints in (6).
>
>
> **Q6: How can the gating model assign the correct weight to the LoRA output when the input distribution is similar across the tasks, while the output is different?**
>
> **A6** As mentioned in **Q2** and **A2**, when the same input is used for different types of tasks (e.g., summarization vs. classification), our gating model leverages the instructions to assign the correct weight to the LoRA output.
>
> For tasks within the same type (e.g., classification tasks), if the input is exactly the same, they can essentially be regarded as the same task. In practice, tasks within the same type typically differ in domain and style. With these differences in input distribution, our gating model can assign the correct weight to the LoRA output.
>
> [1] Wang X, Chen T, Ge Q, et al. Orthogonal Subspace Learning for Language Model Continual Learning, EMNLP 2023.
>
> [2] Zhao W, Wang S, Hu Y, et al. SAPT: A Shared Attention Framework for Parameter-Efficient Continual Learning of Large Language Models, ACL 2024.

---

> > ### Comment · Reviewer_5sqb · 2025-08-06
> >
> > Thank you for the response and the clarifications. Based on the response, I will raise my score.

---

### Official Review · Reviewer_KRVe · 2025-07-05

**Clarity:** 3
**Significance:** 3
**Originality:** 3
**Rating:** 5
**Confidence:** 3

**Summary:**

This paper addresses the challenge of parameter-efficient fine-tuning in multi-task settings, where naively combining task-specific LoRA adapters often leads to performance degradation. It introduces a gated integration framework called GLoRA, which enables dynamic weighting over pre-trained LoRA experts using a lightweight gating module. GLoRA can be plugged into existing PEFT workflows and improves both multi-task and continual learning performance, without requiring retraining or task-specific annotations.

**Questions:**

none

**Ethical Concerns:**

["NO or VERY MINOR ethics concerns only"]

**Limitations:**

yes

**Quality:**

3

**Strengths And Weaknesses:**

This paper tackles the pain point that naïve fusion of task-specific LoRA adapters hurts multi-task performance, and puts forward GLoRA—a gated integration scheme that
(1) keeps the LoRA parameter budget tiny,
(2) lets a lightweight gate dynamically weight experts at run-time, and
(3) plugs straight into existing PEFT pipelines without re-training the backbone. The authors back the idea with broad experiments on ten diverse NLP tasks, three foundation models (T5-Base, OPT-1.3B, LLaMA2-7B), plus ablations on gate order and continual-learning settings, demonstrating consistent gains and limited forgetting. Implementation is simple (just an extra MLP and softmax), code-path changes are minimal, and the paper provides clear analyses of parameter/compute overhead.

---

> ### Author Rebuttal · Authors · 2025-07-31
>
> Thank you for the positive and encouraging feedback. We appreciate your recognition of the motivation and effectiveness of our method.

---

### Note · Authors · 2025-08-13

We appreciate the reviewers’ recognition of GainLoRA’s technical soundness, empirical strength, and practical applicability, and note that two reviewers explicitly committed to increase their ratings after the rebuttal.

**Strengths recognized by reviewers:**

1.	Addresses a key pain point in LoRA-based continual learning: Naïve fusion of task-specific LoRA adapters can hurt performance. (Reviewer KRVe)

2.	Innovative gating mechanism with orthogonal constraints: Proposes an innovative gating module for integrating new and old LoRA branches. (Reviewer fYm8)

3.	High compatibility and general applicability: GainLoRA method has good compatibility (Reviewer fYm8). The proposed GainLoRA can be plugged into other existing LoRA-based methods (Reviewer uWep).

4.	Extensive experimental validation: the idea with broad experiments (Reviewer KRVe); The ablation studies validate the importance of the different components (Reviewer 5sqb); The experiments and ablations are comprehensive (Reviewer uWep).

5.	Clear presentation and intuitive explanation: The flow of the paper is easy to follow (Reviewer 5sqb). The presentation is straightforward and easy to follow (Reviewer uWep).

6.	Directly tackles an important CL challenge for LMs: The paper discusses the problem of catastrophic forgetting in LMs in the CL setting, which is a critical (Reviewer 5sqb).

**Reviewer-specific concerns and resolutions:**

Reviewer R5sqb raised concerns about the lack of explanation for (9) and (10), and task identification from inputs. We provided a detailed mathematical proof and clarified that our method relies on instruction-tuning for task identification. Reviewer R5sqb acknowledged our responses and explicitly confirmed to raise score.

Reviewer uWep raised concerns about novelty, outputs for previous tasks, and prompt-based baseline reimplementation. We clarified our novelty, reported gating outputs for all tasks, and detailed the baseline reimplementation. The reviewer stated that most concerns were resolved and explicitly committed to raise their score.

Reviewer fYm8 raised concerns about newer models, potential loss of useful old-task information due to orthogonality constraints, and overlapping-task scenarios. We added Qwen-3-8B and LLaMA-3-8B results, analyzed similar-task cases, and discussed overlapping-task scenarios. Reviewer fYm8 acknowledged our responses and maintained the positive score.

---

### Decision · Program_Chairs · 2025-09-17

**Decision:**

Accept (poster)

**Comment:**

**Summary**

GainLoRA presents a parameter-efficient continual learning framework that addresses catastrophic forgetting in LoRA-based multi-task settings. The method introduces a gated integration mechanism where each new task receives a dedicated LoRA branch, and lightweight gating modules dynamically weight the contributions of different LoRA branches. The core technical innovation lies in orthogonal constraints applied to both initialization and updates of gating modules, ensuring that new task learning minimally interferes with previously acquired knowledge.

**Reason to accept**
- The paper tackles the important challenge of catastrophic forgetting in LoRA-based continual learning, where naive fusion of task-specific adapters degrades multi-task performance—a recognized pain point in the field.

- The orthogonal constraint mechanism for both initialization and updates of gating modules represents a novel approach to ensuring minimal interference between tasks, going beyond existing gating mechanisms in the literature.

- Multiple reviewers praised the paper's clarity and intuitive explanation of the methodology, making it accessible and easy to follow.

**Summarize the discussion**
Author provide good rebuttal:
- Reviewer **5sqb** initially raised concerns about mathematical explanations and task identification mechanisms. The authors provided detailed mathematical proofs for orthogonal projection operations and clarified their instruction-tuning approach for task identification. The reviewer acknowledged these responses and explicitly confirmed a score increase.

- Reviewer **uWep** questioned novelty, gating outputs analysis, and baseline reimplementation issues. The authors clarified their technical novelty through orthogonal constraints, provided comprehensive gating output statistics across all tasks, and corrected implementation issues with prompt-based baselines. The reviewer stated most concerns were resolved and committed to raising their score.

- Reviewer **fYm8** requested evaluation on newer models and raised concerns about orthogonal constraints potentially limiting knowledge transfer. The authors added Qwen-3-8B and LLaMA-3-8B results and provided insightful analysis of the forward transfer trade-off using similar tasks, acknowledging this as a fundamental continual learning challenge.

- Reviewer **KRVe** provided consistently positive feedback throughout, recognizing the method's effectiveness and practical value.
The authors' responses demonstrated deep understanding of their method's limitations while providing substantial additional evidence and clarifications that strengthened the submission.